# Top-down modulation of sensory processing and mismatch in the mouse posterior parietal cortex

Constanze Raltschev[1,3], Sergej Kasavica[1,3], Benjamin Leonardon [1], Thomas Nevian[1] & Shankar Sachidhanandam [1,2] ✉

An important function of the neocortex is to compare sensory feedback stimuli with internal predictions of the outside world and evoke mismatch responses to deviations, thus allowing expectations to be updated. The mechanisms behind sensory feedback mismatch and prediction formation however remain unclear. Here we created a learned association of an auditory-tactile stimulus sequence in awake head-fixed mice, where a sound predicted an up-coming whisker stimulus and introduced mismatches by omitting or altering the whisker stimulus intensity. We showed that layer 2/3 posterior parietal cortex (PPC) neurons could report stimulus sequence mismatches, as well as display neural correlates of expectation. Inhibition of PPC-projecting secondary motor cortex (M2) neurons suppressed these correlates, along with population mismatch responses. Hence, M2 can influence sensory processing in the PPC and potentially contribute to the prediction of sensory feedback from learned relationships within sequences of sensory stimuli.

The experience of different sensory stimuli enables us to make associations between them, especially when presented as part of a sensory sequence. For example, we learn to anticipate the sound of thunder after a flash of lightning. We would be surprised in the absence of the expected stimulus because there was a mismatch between what we anticipated and what we actually experienced. The predictive processing framework postulates that the brain constantly predicts sensory feedback based on an internal representation of the outside world built upon prior sensory experience[1,2]. Mismatch responses, or prediction errors, that are generated during deviations between what was predicted and the actual sensory input are then used for updating stimulus expectations[3]. This is an essential feature at both the physiological and behavioral level as it facilitates the detection of unexpected, potentially dangerous events[4], and is thus crucial to survival. Sensory feedback mismatch responses have been described in the primary visual cortex (V1) in mice[5,6], as well as in the primate primary auditory cortex[7] and the primary auditory pallium of songbirds[8]. In mice, local cortical circuits implicated in generating mismatch

responses have been investigated mainly during locomotion in a virtual reality setting. There, the predicted sensory feedback from self-generated motion was compared against the actual sensory flow, when coupled with closed-loop visual, auditory, or tactile stimuli[9–11]. Conversely, it was recently shown that neurons within the posterior parietal cortex (PPC) of awake head-fixed mice can report mismatches in sensory sequences in the absence of locomotion/motor-output-related sensory feedback predictions, such as during the omission of a previously experienced passive touch to the whiskers[12]. The PPC, a sensory associative area, receives bottom-up sensory input from the primary sensory cortices[13–15] and has reciprocal connections with the higher order cortical areas (secondary motor cortex M2, anterior cingulate cortex A24, midcingulate cortex A24', and orbitofrontal cortex) that can provide top-down feedback information[16–18]. It could therefore compare incoming sensory stimuli with the expectations of these stimuli constructed from prior experience, and report mismatches between the two, analogous to what has been described in visual processing in mouse V1[19]. However, little is known about how such

[1]Department of Physiology, University of Bern, Bern, Switzerland. [2]Present address: Laboratory of Sensory processing, Faculty of Life Sciences, Brain Mind Institute, École Polytechnique Fédérale de Lausanne (EPFL), Lausanne, Switzerland. [3]These authors contributed equally: Constanze Raltschev, Sergej Kasavica. ✉e-mail: shankar.sachidhanandam@unibe.ch

learned associations formed from experiencing sensory sequences, where one stimulus predicts another, can influence sensory processing as well as the generation of mismatch responses.

Here, we presented awake head-fixed mice with an auditory stimulus via a loudspeaker followed by tactile stimuli through whisker deflection with a magnetic coil. Hence, we created a sensory stimulus sequence where a sound predicts the upcoming whisker stimulus. We could then introduce a mismatch in this sequence by varying the intensity of the whisker stimuli, or omitting its presentation while recording neuronal activity of layer 2/3 PPC neurons at single-cell resolution with in vivo 2-photon calcium imaging. We demonstrate that the PPC can represent associations of sequences of sensory stimuli and reliably generate mismatch responses to deviations in these sequences. We further reveal that this dynamic representation of sensory stimuli is modulated by top-down feedback from M2 and can be interpreted within the framework of predictive processing.

## Results

### Creating a sensory association at the PPC and reporting mismatch

To create an association between two sensory stimuli, we first presented awake head-fixed mice with an auditory tone ("sound" session), either looming or non-looming in intensity (Fig. 1a). This was followed by trials with only a whisker stimulus ("whisker" session). We then presented the whisker stimulus immediately after the auditory stimulus ("pairing" session), in this case, the looming sound, so that the sound would predict the upcoming whisker stimulus. In subsequent trials ("interleaved" session), we created a mismatch (mismatch trials) in the experienced auditory-tactile sequence by randomly omitting the whisker stimulus in 20% of the trials. Hence the interleaved session comprised of randomized matched (sound followed by whisker stimulus) and mismatch (sound followed by whisker stimulus omission) trials (Fig. 1a). These sessions were presented sequentially within the same imaging session. Neuronal activity within the PPC was measured using 2-photon calcium imaging at single-cell resolution in layer 2/3 neurons of mice expressing the genetically encoded calcium indicator (GECI) RCaMP1.07[20], with a chronic cranial window in place. Expression of this viral construct was previously shown to be mainly in pyramidal neurons[11]. Using intrinsic optical signal (IOS) imaging (see Methods and Supplementary Fig. 1) we localized area A of the PPC via exclusion, where mismatch responses to the omission of tactile stimuli were previously reported[12].

We first identified neurons (6 mice, 1 field of view, FOV, per mouse) that responded to one of the presented stimuli (sound or whisker stimulus) in any of the four sequentially presented sessions. We then classified them as significantly responsive when compared to their respective shuffled data to account for random noise (see Methods). We subsequently based all our analysis on these shuffle-corrected neurons and followed their activity through the different sessions described below. Here, we present the mean ΔF/F computed during a 1 s window from stimulus onset minus the baseline mean ΔF/F 1 s prior to stimulus. We focus primarily on the whisker stimulus window responsive neurons in the whisker, pairing, and interleaved sessions, and describe the sound-responsive neurons in Supplementary Fig. 2. During whisker stimulation ("whisker" session) 39.4% of PPC neurons (329 of 834 neurons, 37.3% response probability) were responsive to whisker deflection (Fig. 1d, whisker-responsive neurons). The pairing of auditory and whisker stimuli ("pairing" session) recruited a new population of neurons that were previously weakly or non-responsive to the whisker stimulus alone (Fig. 1d, pairing-responsive neurons). Whisker stimulus-evoked responses were larger in the pairing sessions compared to whisker sessions ($p = 7.9 \times 10^{-4}$, 329 whisker-responsive neurons, 154 pairing-responsive neurons, with 33 overlapping neurons or 7.3%, Wilcoxon-Mann-Whitney test) (Fig. 1e).

In the subsequent interleaved session, we found that 8.5% of neurons within the PPC (70 of 834 neurons, 32.1% response probability) could report the omission of the previously experienced whisker stimulus (Fig. 1d, mismatch-responsive neurons). These mismatch responses were observed from the very first mismatch trial and were overall stable over time (Fig. 1f). Among the mismatch-responsive neurons, only three neurons were previously identified as looming offset-responsive in the sound session. This confirms that the mismatch neurons represent a newly recruited group of cells that report whisker stimulus omission and are not merely sound offset-responsive neurons. The mismatch response was also not associated with whisker movements (Fig. 1b, d, interleaved mismatch), consistent with previous studies[12]. We also observed mismatch responses of comparable size when the non-looming sound was paired with the whisker stimulus (Supplementary Fig. 2). In the interleaved session, we found that a larger fraction of neurons (277 of 834, 33.2%, with 31.7% response probability) were responsive to the interleaved matched trials, but with a significantly smaller response compared to the mismatch response ($p = 0.008$, 70 mismatch-responsive neurons, 277 interleaved matched-responsive neurons, Wilcoxon-Mann-Whitney test). Also, the interleaved matched-responsive neurons in the PPC displayed an increase in the mean ΔF/F 0.5 s before whisker stimulus onset (shaded window in Fig. 1d) from the pairing to interleaved session. This pre-stimulus increase was absent in the pairing-responsive neurons (Fig. 1g). Hence, the PPC can represent sensory stimuli in a stimulus sequence and report mismatches in the experienced sequence, with mismatch responses being larger than that of the predicted stimulus.

### Neural correlates of expectation in the PPC

We then questioned if the pre-stimulus increase in the mean ΔF/F during the predicting sound window (Fig. 1g) could represent a neural correlate of expectation. In a separate set of experiments, we presented mice (6 mice, 6 FOV in total) to a first interleaved session where the whisker stimulus was delivered immediately after the sound (as in Fig. 1a). This was followed immediately by a second interleaved session where we introduced a 1 s delay between the sound and the onset of the whisker stimulus, to test if the pre-stimulus increase in neural response could be prolonged and/or enhanced. We observed that in the interleaved matched trials, the pre-stimulus response was indeed prolonged as well as enhanced with the delay, along with a decrease in the post-stimulus whisker response (Fig. 2a, b). In the mismatch trials, there was no corresponding prolongation, and the pre-stimulus response was comparable to that without the delay. The mismatch responses in both sessions were also comparable in size (Fig. 2c, d). Hence the pre-stimulus response in the interleaved matched-responsive neurons could potentially represent a top-down driven neural correlate of expectation in the PPC, as mismatch trials are interleaved with the matched trials.

### PPC can report different types of sensory mismatch

So far, we employed the omission of the whisker stimulus as the mismatch in the interleaved trials. Can the PPC also report mismatches in stimulus intensity? We designed two different types of interleaved sessions that were presented separately to the mice (Fig. 2e). In one session, we interleaved the matched trials with mismatch trials that had a smaller whisker stimulus intensity. These decreased-intensity trials would then signal a negative mismatch. In contrast, to signal a positive mismatch in another session, we interleaved the matched trials with mismatch trials of a higher whisker stimulus intensity (increased-intensity trials). This would allow for the identification of potential negative and positive mismatch neurons as described in the canonical microcircuit for predictive processing[21]. Hence three separate interleaved sessions with either omission, decreased or increased whisker stimulus intensities as a mismatch were performed

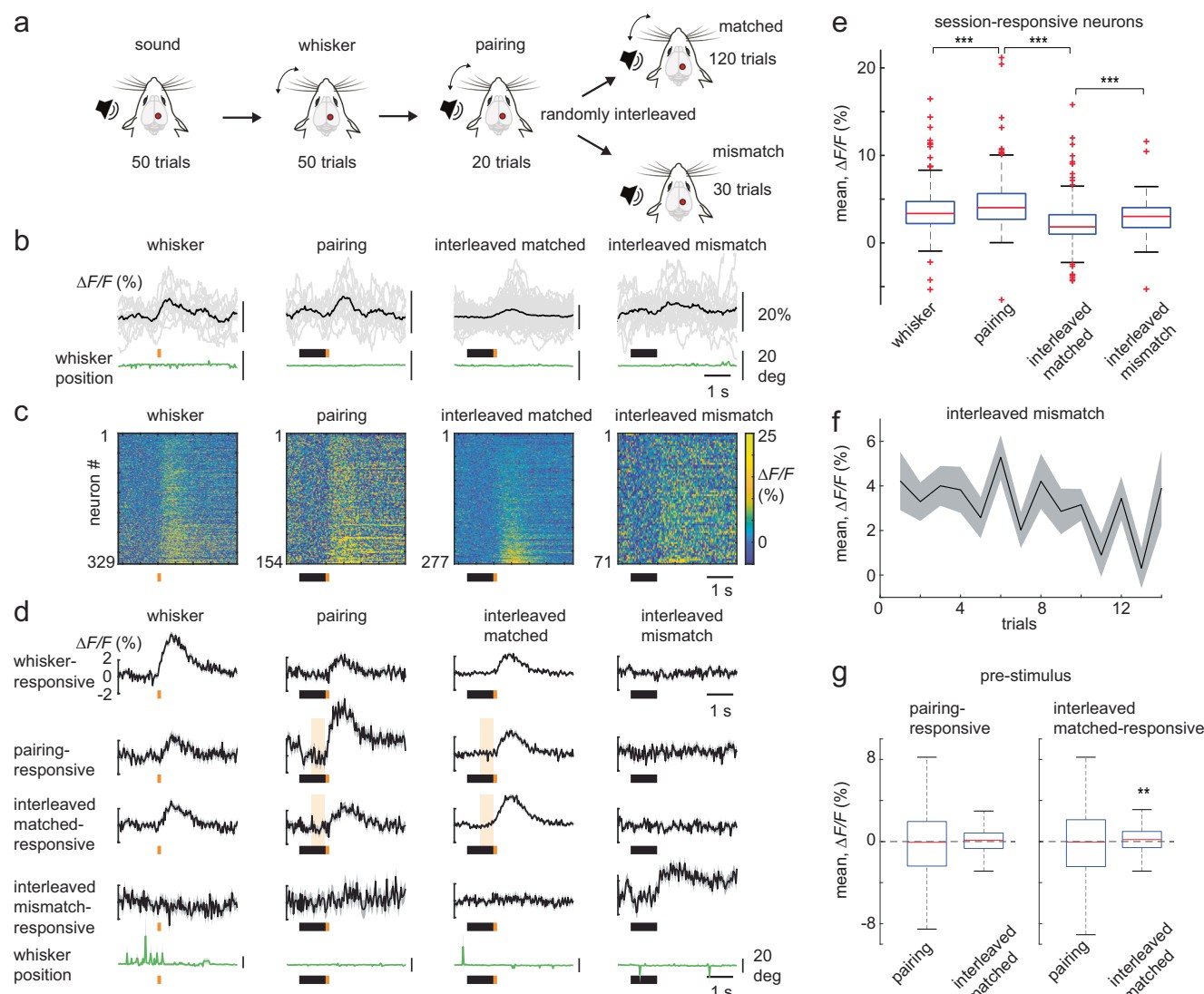

**Fig. 1 | Creating a sensory association at the PPC. a** Schematic of the sequence of stimulus presentation carried out in a single imaging session. All trials begin with a brief sound cue (not shown). Looming and non-looming sounds are randomly interleaved in the sound trials. The whisker stimulus (orange dot in (**b**)) delivered by a magnetic coil is presented alone in the whisker session. The looming sound (black bar in (**b**)) is paired with the whisker stimulus in pairing session. In the interleaved session, the matched trials (looming sound followed by whisker stimulus) are randomly interleaved with the mismatch trials where the whisker stimulus is omitted. **b** Example stimulus-responsive neurons classified as whisker-, pairing-, interleaved matched-, and interleaved mismatch-responsive neurons from layer 2/3 of the PPC, expressing RCaMP1.07. Average (black) and single trial (grey) ΔF/F calcium traces are shown, along with the corresponding average whisker position (green). **c** Heat maps representing the average stimulus responses of whisker- (329 neurons), pairing- (154 neurons), interleaved matched- (277 neurons), and interleaved mismatch-responsive neurons (70 neurons) from 6 wild-type mice. Mean pre-stimulus activity (1 s) was subtracted from the calcium transients and neurons were sorted to their mean ΔF/F response times in the stimulus

window. **d** Population averages (± s.e.m.) of ΔF/F traces of the neurons in (**c**), along with their corresponding population average for the other presented stimuli. The average whisker position from the above sessions is shown below in green. **e** Boxplot of average population responses of the whisker-, pairing-, matched- and mismatch-responsive neurons shown in (**c**). **f** Average population responses of mismatch-responsive neurons over mismatch trials. **g** Boxplot of pre-whisker stimulus-response of the pairing- and interleaved matched-responsive neurons in (**c**), during their pairing and interleaved-matched sessions. Pre-stimulus response corresponds to shaded area in (**d**). Outliers are shown in Supplementary Fig. 2f. Boxplot central line indicates the median, the bottom and top edges of the box indicate the 25th and 75th percentiles respectively, the whiskers extend to maximum and minimum points within 1.5 s.d., and outliers are marked with crosses (**e**, **g**). Data are represented as mean ± s.e.m. in (**f**). Statistical significance is indicated by ** for $p < 0.01$ and *** for $p < 0.001$, with two-sided Wilcoxon-Mann-Whitney test in (**e**) and two-sided Wilcoxon sign-rank test in **g**. The mouse illustration in panel **a** was adapted from an image created in the laboratory of Carl Petersen, EPFL.

sequentially in a single imaging session. The PPC was able to effectively report mismatches in stimulus intensity in both directions, where the mismatch response was larger than the corresponding interleaved matched response (Fig. 2g, h). The mismatch responses were also encoded by largely separate groups of neurons (Fig. 2f and Supplemental Fig. 3b), indicating the presence of distinct negative and positive mismatch neurons. We further observed that the pairing-

responsive neurons showed a response in the increased-intensity mismatch trials, although comparable in size to their interleaved matched trial response (Supplemental Fig. 3c, d). Conversely the increased-intensity mismatch neurons were responsive in the prior pairing trials. Both groups displayed smaller responses to the whisker stimulus in the interleaved session with the omission-mismatch trials (Supplementary Fig. 3c, d). This suggests that the positive mismatch

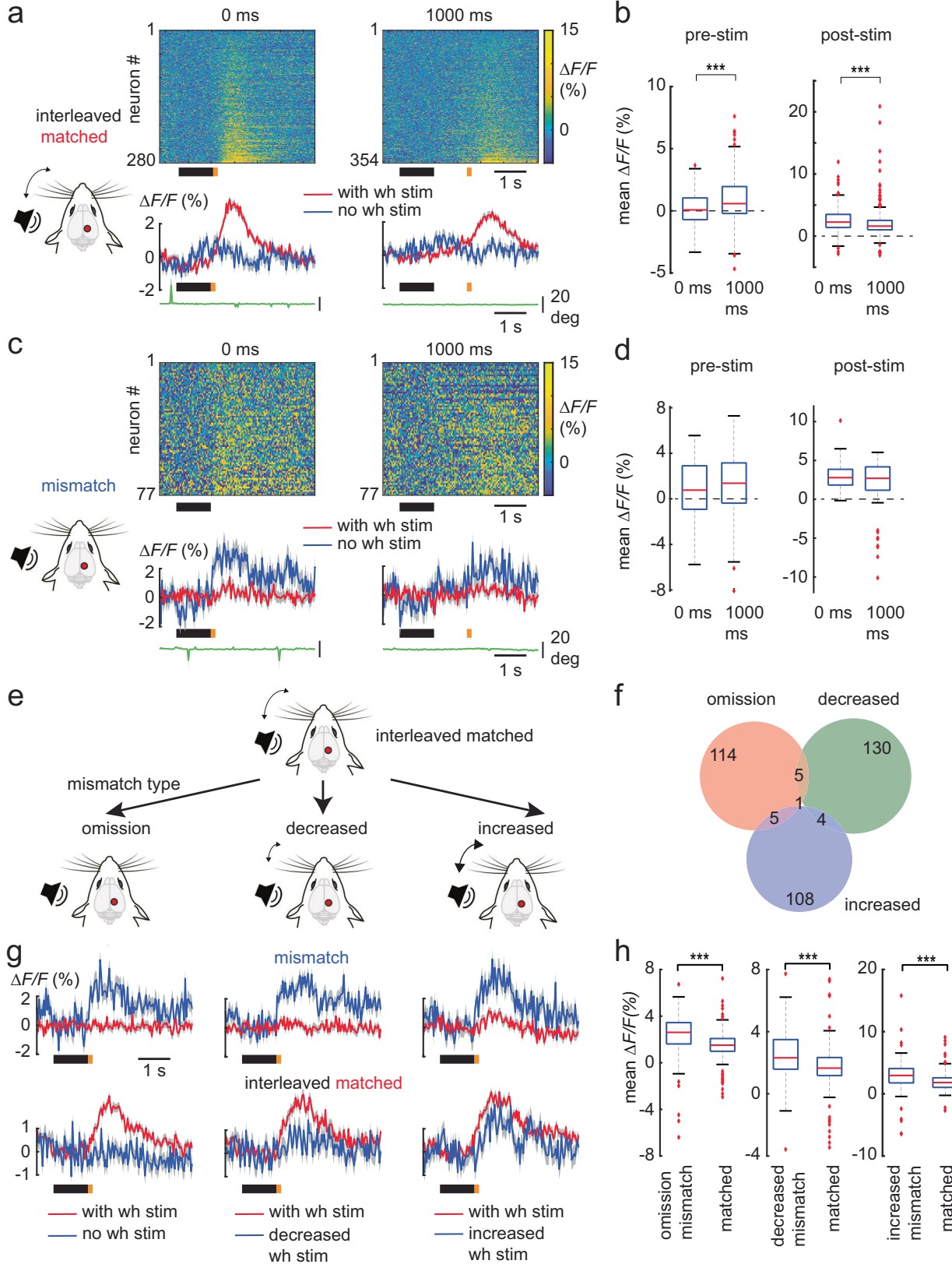

neurons may be part of a subgroup within the pairing-responsive neurons (see Discussion). Thus, the PPC can report positive and negative mismatches with distinct groups of neurons whose response dynamics can match the canonical microcircuit for predictive processing.

## M2 modulates sensory processing in the PPC

Next, we investigated the source of the top-down feedback to the PPC that could drive the pre-stimulus response/expectation. In addition to the primary sensory cortices, the PPC has strong reciprocal connections with higher-order cortical areas, including M2[12,18]. These cortices

**Fig. 2 | PPC can reliably update the mismatch response and report different types of mismatch. a** Heat maps and corresponding population averages (± s.e.m.) of ΔF/F traces of interleaved matched-responsive neurons with no delay (0 ms) and a 1000 ms delay between the looming sound and whisker stimulus presentation (without delay, 280 neurons; with delay, 354 neurons; 6 mice; 6 FOVs). The average whisker position from the above sessions is shown below in green. **b** Boxplot of average population pre-stimulus and post-stimulus responses of the interleaved matched-responsive neurons in (**a**), without, and with delay. **c, d** same as (**a, b**), but for mismatch-responsive neurons (without delay, 77 neurons; with delay, 77 neurons). **e** Schematic of the presentation of different mismatch trial types in the form of whisker stimulus omission, decreased and increased whisker stimulus intensity. **f** Venn diagram of the different mismatch-responsive neurons in (**g**), their respective overlaps between mismatch sessions. **g** Population averages (± s.e.m.) of ΔF/F traces of interleaved mismatch-responsive neurons and matched-responsive

neurons, for the interleaved sessions with omission (125 mismatch neurons, 313 matched neurons), decreased (140 mismatch neurons, 221 matched neurons) and increased (118 mismatch neurons, 314 matched neurons) whisker stimulus mismatches, with their corresponding matched (red) and mismatch (blue) trial averages. **h** Boxplot of average population responses of the mismatch-responsive and matched-responsive neurons in (**g**), for the interleaved sessions with omission, decreased and increased whisker stimulus mismatches. wh stim: whisker stimulus. Boxplot central line indicates the median, the bottom and top edges of the box indicate the 25th and 75th percentiles respectively, the whiskers extend to maximum and minimum points within 1.5 s.d., and outliers are marked with crosses (**b, d, h**). Data are represented as mean ± s.e.m. Statistical significance is indicated by *** for $p < 0.001$, with two-sided Wilcoxon-Mann-Whitney test. The mouse illustrations in (**a, c,** and **e**) were adapted from an image created in the laboratory of Carl Petersen, EPFL.

have been shown to exchange sensory and motor information, respectively[22], with the PPC. To test the potential contribution of M2 as the source of top-down feedback, we expressed the inhibitory DREADD hM4Di in ipsilateral PPC-projecting M2 neurons using a retro-Cre-based viral expression strategy in mice that expressed RCaMP1.07 in PPC neurons (3 mice, 6 FOVs with and without CNO) (Fig. 3a and see Methods). We then administered CNO intraperitoneally (i.p.) to the mice to inhibit the activity of the PPC-projecting M2 neurons and presented the auditory-tactile sequence as described in Fig. 1a. We compared their neuronal response to that of control mice that were similarly administered CNO i.p. and did not express the inhibitory DREADD (4 mice, 6 FOVs). We observed that with the reduction of top-down feedback from ipsilateral M2 (hM4Di+CNO), the whisker stimulus responses of the pairing-responsive neurons (Fig. 3b, c) and that of the interleaved matched-responsive neurons (Fig. 3d, e) remained unchanged. The reduction of M2 feedback however reduced the pre-stimulus response of the interleaved matched-responsive neurons in the interleaved matched session, compared to the +CNO control experiments (Fig. 3d, f). For the interleaved mismatch trials, we found that neurons could still report the omission mismatch, with a response size comparable to +CNO control experiments in the absence of M2 suppression (Fig. 4a, b). There were, however, less neurons recruited to report the mismatch when M2 was inhibited (4.1%, 42/1014 vs 8.8%, 78/882 for hM4Di+CNO and +CNO control experiments respectively). At the population level, the interleaved mismatch trials displayed a response to the sound that predicted the upcoming whisker stimulus, followed by a mismatch response to the omission of the whisker stimulus in the +CNO control experiments (Fig. 4c). When we analyzed the impact of M2 inhibition on the population mismatch trials, we observed that the response to the predicting sound as well the mismatch response was greatly suppressed compared to that in the +CNO control experiments (Fig. 4c, d). Taken together, these findings show that M2 can modulate sensory processing in the PPC, driving the pre-stimulus response/expectancy in different groups of neurons. Furthermore, M2 can potentially contribute to the top-down prediction arriving at the PPC that modulates the response to the predicting sound, as well as the omission mismatch response (Supplementary Fig. 4).

## Discussion

Using an auditory-tactile stimulus sequence, we were able to follow the formation of a sensory association in mice, along with the updating of stimulus expectations while we introduced mismatches in the sequence, as represented by the neural activity of layer 2/3 neurons in the PPC. We have shown that the PPC is able to dynamically represent sensory sequences with different groups of neurons. One of the features of learned sensory associations is the inhibition of the expected bottom-up input[23,24]. This became evident in the size of the whisker stimulus-response, which was significantly reduced in the transition from pairing to interleaved trials. Indeed, both the whisker- and

pairing-responsive neurons were suppressed outside of their respective sessions. Top-down feedback can suppress[25], as well as enhance[26–28] sensory processing in primary sensory cortices, as part of a widespread cortical circuit motif[26,29]. As parvalbumin (PV), vasoactive intestinal peptide (VIP), and somatostatin (SST) expressing inhibitory neurons can be directly activated by top-down inputs[25,26,30], it would be important to determine their role in our findings.

In our experiments, we used a sound presentation to predict an up-coming whisker stimulus, and subsequently show the appearance of a pre-stimulus response that could represent a neural correlate of expectation (Fig. 1d, g). This pre-stimulus response was present in the interleaved matched-responsive neurons and could be prolonged and enhanced temporally upon introducing a delay between the predicting sound and the expected whisker stimulus (Fig. 2a, b). We were able to suppress the pre-stimulus response with the inhibition of M2 feedback to the PPC (Fig. 3d, f), demonstrating that M2 can potentially contribute in part to the top-down expectation/prediction to the PPC, which could also be interpreted as an attentional signal[26]. Neurons in the PPC can then compare this prediction with the actual bottom-up sensory information and report mismatches.

We have demonstrated that the PPC can report mismatches in the form of omission as well as deviations from the predicted strength of the whisker stimulus. Based on the canonical microcircuit for predictive coding[10,21], these mismatch neurons could be categorized into negative and positive error neurons that would report mismatches that are less (omission and decreased intensity mismatch) or more (increased intensity mismatch) than what is expected. In our experimental paradigm, the positive error neurons could form part of the pairing-responsive neurons, as they become suppressed by top-down feedback from M2 (Supplementary Fig. 4), as the prediction is formed in the interleaved sessions. Indeed, when the bottom-up input exceeded this prediction, these neurons could report this positive mismatch (Supplementary Fig. 3b). The matched-responsive neurons, on the other hand, showed mixed responses, integrating both top-down predictions and bottom-up stimulus information, in the form of the pre- and post-stimulus response. Finally, the omission and decreased-intensity mismatch-responsive neurons could function as negative error neurons as they are driven largely by the top-down prediction. Notably, we observed little overlap between the omission and decreased-intensity mismatch neurons, with mismatch responses of comparable size. Differences in the degree of bottom-up driven inhibition for the two mismatch types could potentially level out eventual expected differences in prediction error magnitude. It should be noted that the fraction of mismatch-reporting neurons in our experiments is small, compared to that observed in V1[10] for sensorimotor mismatches (approximately 10% vs 40%). Nonetheless, the mismatch responses in the PPC were larger than the corresponding matched responses in the interleaved sessions. Hence there are neurons within the PPC that can contribute to reporting mismatches in sensory sequences.

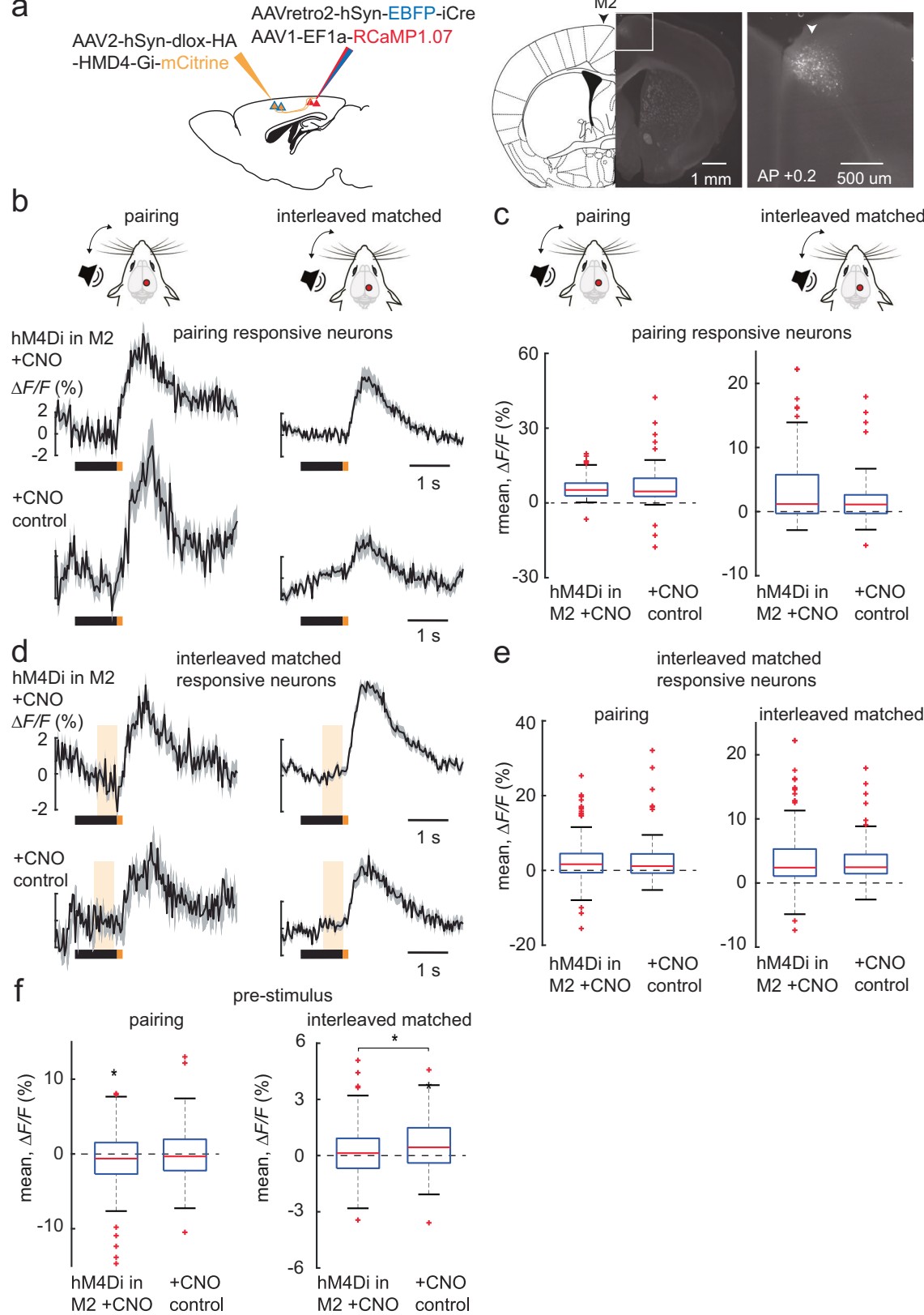

While the silencing of ipsilateral M2 feedback to the PPC reduced the number of mismatch neurons, it did not affect the size of single-neuron mismatch responses. This suggests that multiple higher-order areas, in addition to ipsilateral M2, could contribute to the top-down prediction. Nonetheless, the mismatch response was suppressed at the net population average. This was also accompanied with the

suppression of the sound response that predicted the whisker stimulus (Fig. 4c). Sound-responsive neurons in the sound session (done before the whisker and pairing sessions in Fig. 1a) did not respond to the predicting sound in the interleaved session (Supplementary Fig. 2b). This was unlike the neurons that responded to the whisker stimulus in the whisker, pairing and interleaved matched trials, that continued to

**Fig. 3 | M2 modulates sensory processing in the PPC. a** Injection scheme to express the inhibitory DREADD hM4Di in M2 neurons projecting to the PPC (sagittal view), along with a coronal section showing their expression in M2 neurons as represented by mCitrine fluorescence. **b** Population averages (± s.e.m.) of ΔF/F traces of pairing-responsive neurons, in the pairing and interleaved matched sessions, for the hM4Di+CNO (3 mice, 6 FOVs, 79 neurons) and +CNO control experiments (4 mice, 6 FOVs, 52 neurons). **c** Box plot of post-whisker stimulus responses of the pairing-responsive neurons in (**b**), for their pairing and interleaved matched sessions. **d** Population averages (± s.e.m.) of ΔF/F traces of interleaved matched-responsive neurons, in the pairing and interleaved matched sessions, for the hM4Di+CNO (206 neurons) and +CNO control experiments (90 neurons). **e** Boxplot of post-whisker stimulus responses of the interleaved matched-

responsive neurons in **d**, for their pairing and interleaved matched sessions. **f** Boxplot of average population pre-stimulus responses of the interleaved matched-responsive neurons in (**d**), for their pairing and interleaved matched sessions. Boxplot central line indicates the median, the bottom and top edges of the box indicate the 25th and 75th percentiles respectively, the whiskers extend to maximum and minimum points within 1.5 s.d., and outliers are marked with crosses (**c**, **e**, **f**). Data are represented as mean ± s.e.m. Statistical significance is indicated by * for *p* < 0.05 with two-sided Wilcoxon-Mann-Whitney test and two-sided Wilcoxon sign-rank test. The schematic drawings of the brain in (**a**) are reproduced from Paxinos and Franklin (2001)[35] with permission from Elsevier. The mouse illustration in (**b**) was adapted from an image created in the laboratory of Carl Petersen, EPFL.

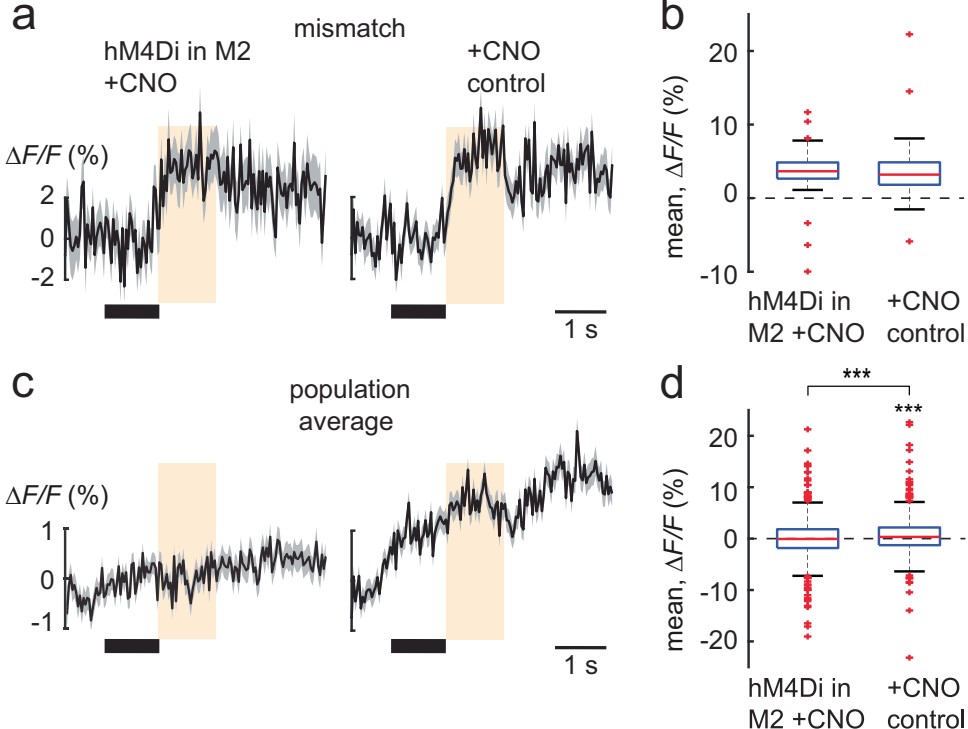

**Fig. 4 | M2 contributes to the top-down prediction to the PPC. a** Population averages (± s.e.m.) of ΔF/F traces of interleaved mismatch-responsive neurons in layer 2/3 neurons in the PPC with M2 suppression using hM4Di+CNO (41neurons) and with +CNO control experiments (75 neurons) (same mice as in this figure). **b** Boxplot of average population responses of the interleaved mismatch-responsive neurons in (**a**). **c** Population averages (± s.e.m.) of ΔF/F traces of all neurons during the mismatch trials in hM4Di+CNO (*n* = 1014 neurons) and +CNO

control experiments (*n* = 882 neurons). **d** Boxplot of average population responses of the neurons in (**c**) during the mismatch trials. Boxplot central line indicates the median, the bottom and top edges of the box indicate the 25th and 75th percentiles respectively, the whiskers extend to maximum and minimum points within 1.5 s.d., and outliers are marked with crosses (**b**, **d**). Data are represented as mean ± s.e.m. Statistical significance is indicated by *** for *p* < 0.001, with two-sided Wilcoxon-Mann-Whitney test and two-sided Wilcoxon signed-rank test.

respond across the sessions (Fig. 1d). Hence it is possible that M2 feedback modulates this response to the predicting sound as well, either directly[27] or indirectly via disinhibition[26], once the sensory association has been learned. This demonstrates a strong effect of M2 on the population code.

Based on these findings, we propose that during the interleaved matched trials, top-down inputs predicting the whisker stimulus are matched by bottom-up whisker stimuli that recruit neighboring inhibitory neurons, such as the SST inhibitory neurons. These neurons are strongly driven by local excitation and could inhibit the dendrites of layer 2/3 neurons in the PPC[10,21]. In the interleaved mismatch trials, the top-down prediction can directly activate the dendrites of the PPC neurons in the absence of this locally recruited inhibition to drive the mismatch response. This would be in accordance with the framework of predictive coding during sensorimotor transformations and brings our study in line with the hierarchical visual processing described in mouse V1[10,19].

## Methods

### Experimental model and subject details

All experimental procedures followed the guidelines of the Veterinary Office of Switzerland and were approved by the Cantonal Veterinary Office in Bern. The data was collected from wild-type C57/BL6 mice (*n* = 15) and both males and females were used. All mice were at least 8 weeks old at the time of viral injection/head-post implantation.

### Experimental design

This study did not involve randomization or blinding. No data or mice were excluded from the analysis.

### Surgery

Mice were anesthetized with isoflurane (3%) and subcutaneously injected with carprofen (5 mg/kg) prior to surgery for viral injections. For calcium imaging, the genetically encoded calcium indicator (GECI) RCaMP1.07 was injected into the PPC at 1.7 mm lateral and 2.0 mm

posterior of bregma, to target layer 2/3 (~250 μm below the pial surface). For the silencing of M2 neurons, the DREADD hMD4i was injected into M2 at 0.5 mm lateral and 0.2 mm anterior of bregma, to target layers 2/3 and 5 (~250 and 500 μm below the pial surface). For long-term in vivo calcium imaging, a cranial window was implanted 24 h to 1 week after virus injections over the PPC as described previously[31]. Mice were anesthetized with isoflurane and subcutaneously injected with buprenorphine (0.1 mg/kg) prior to surgery for window implantation. Briefly, a craniotomy was performed at the injection site. A cover glass (4 mm diameter) was placed directly over the exposed dura mater and sealed to the skull with dental acrylic. A metal post was fixed to the skull with dental acrylic, posterior to the cranial window, to allow for subsequent head fixation. One week after chronic window implantation mice were handled daily for another week, and gradually habituated to head fixation. Intrinsic optical signal (IOS) imaging was performed on the mice to identify the location of PPC by exclusion as previously described[12]. In brief, to avoid the activation of surrounding whiskers, all whiskers except the γ-barrel-column whisker on the right whisker pad of the mice were trimmed and their locations were mapped using IOS on the exposed skull during whisker stimulation (rostrocaudal deflections at 10 Hz). The location of the primary visual cortex (V1) was similarly mapped using full-field stimulation with a green LED placed 5 mm in front of the contralateral eye. The non-activated region between these two identified sites was delineated as PPC for subsequent imaging sessions.

## Viral constructs
For calcium imaging, AAV2/1-hEF1α-RCaMP1.07-WPRE-hGHp(A) (300 nL, ~5.0 × 10^12 μg/mL) was injected into the PPC of wild-type mice targeting layer 2/3 to induce expression of the GECI RCaMP1.07 in neurons. For the silencing of M2 neurons, AAVretro2-hSyn-EBFP-iCre was injected in layer 2/3 of the PPC, along with RCaMP1.07 (for calcium imaging), and AAV2/1-hSyn-dlox-HA-hM4D-Gi-mCitrine was injected in M2 to selectively target PPC projecting M2 neurons.

## DREADD inhibition and CNO control experiments
The inhibitory DREADD hM4Di was used to chemogenetically silence M2 neurons projecting to the PPC. An i.p. injection of clozapine N-oxide (CNO dihydrochloride, 1 mg/kg, Tocris cat. no. 4936), the ligand that activates hM4Di, was done 30 min before presentation of the sensory stimulus sequence. We performed multiple imaging sessions with the presence of CNO first over several days, before repeating them in the absence of CNO at least 48 h after the last CNO session. In the CNO control experiments, wild-type mice without any DREADD expression were injected i.p. with CNO (1 mg/kg) 30 min before presentation of the sensory stimulus sequence.

## Presentation of sound and tactile stimuli
To create a sensory association of the stimulus sequence, each mouse was presented with, in the following order, the sound session (~25 randomly interleaved trials each of the looming and non-looming sound), the whisker session (~50 whisker stimulus trials), the pairing session (approximately 20 trials with looming sound followed by the whisker stimulus), the interleaved session (approximately 120 matched trials of looming sound followed by the whisker stimulus, randomly interleaved with approximately 30 mismatch trials). The looming (increasing intensity) and non-looming (constant intensity) sounds were based on previously recorded soundtracks[12] and recreated in MATLAB, that consisted of a cloud of tones (0.1–8 kHz). Each sound was 1 s long and was delivered via a loudspeaker at 75 dB placed contralateral to the imaging site. We used the looming sound to predict the whisker stimulus in the pairing session as mismatch responses to the absence of a tactile stimulus cued by a looming sound could be generated in the PPC[12]. Mismatch responses could also be evoked in the PPC when the non-looming sound was used in the pairing session (Supplementary

Fig. 2), suggesting that a range of auditory cues could be potentially used in the sensory sequence. Tactile stimuli were delivered as deflections (3 × 10 Hz, 1 ms pulse) to multiple whiskers. This was achieved by attaching small metal particles to the whiskers and subsequently moving them via a brief magnetic field generated by a coil placed beneath the head of the mouse[32]. The whisker stimulus intensity was set to 80% for all experiments except for those performed during the intensity mismatch experiments (Fig. 2e, g). Here, the interleaved matched stimulus was set to 60%, and the decreased and increased intensity mismatch stimuli were set to 40% and 80%, respectively. Each trial began with a 2 s baseline, followed by a brief sound cue (50 ms, 85 dB) to signal trial start to the mouse. In the pairing and interleaved trials, the looming sound was played 1 s after the sound cue and was followed immediately by the whisker stimulus. In a subset of control experiments, the non-looming sound was played instead of the looming sound (see Supplementary Fig. 2). Each trial was 8 to 9 s long and trials were presented with an inter-trial interval of 2 to 5 s, to render them irregular in presentation timing. White noise (65 dB) was played during the entire duration of the imaging session. Sound and whisker stimulus delivery was controlled by custom-written software in C.

## Whisker tracking
The whisker field was illuminated with a 940 nm infrared LED light and movies were acquired at 100 Hz (500 × 500 pixels) using a pixy camera system coupled with an Arduino board to track the position of the whiskers in real time[33].

## Two-photon calcium imaging
We used a custom-built 2-photon microscope controlled by Scan-Image 2019 equipped with a fixed wavelength fiber laser at 1064 nm (Fidelity; Coherent) and a Ti:sapphire laser system (~100-fs laser pulses; Mai Tai BB; Newport Spectra Physics), a water-immersion objective (16×LWDPF, 0.8 NA; Nikon), resonant scan mirrors (model 6210; Cambridge Technology), and a Pockel's Cell (Conoptics) for laser intensity modulation. For calcium imaging, RCaMP1.07 was excited at 1064 nm with the Fidelity. Emitted fluorescence was collected with red (617/73 nm) and green (520/60 nm) emission filters respectively. Images were acquired at 30 Hz with 512 × 512-pixel resolution.

## Histology
After the calcium imaging recordings, mice were deeply anaesthetized by intraperitoneal injection of 80/10 ketamine/xylazine mixture and transcardially perfused with 4% paraformaldehyde (PFA). Brains were removed and post-fixed in PFA for 24–48 h at 4 °C and were subsequently washed in phosphate-buffered saline and sliced at 100 μm. Brain slices were mounted using Mowiol® 4-88 prior to imaging on a LEICA m205 FCA fluorescence stereo microscope to confirm the location of the PPC.

## Quantification and statistical analysis
**Two-photon calcium data processing.** Somatic calcium signals were automatically detected using the Python-based CaImAn analysis pipeline, which performed motion correction, source extraction, and component registration[34]. Raw fluorescence traces ($F$) of the calcium signals are presented here as $\Delta F/F = (F-F_O)/F_O$, where $F_O$ was calculated for each trial as the mean of the 1 s window prior to the sound cue that signaled trial start. The responsiveness of a detected neuron to a given stimulus (sound, sound-offset, whisker, mismatch) was determined by comparing the distribution of its single trial responses (single trial mean $\Delta F/F$ calculated in a 1 s window from stimulus onset minus the mean baseline window 1 s before stimulus onset) against the distribution of 1000 randomly selected events from its same session (random baseline-corrected mean $\Delta F/F$ calculated in a 1 s window as above), hence taking into account the random noise of each neuron. Significance was determined with a two-sided Mann-Whitney-U test ($p < 0.05$). We

subsequently classified these neurons as looming-, non-looming-, looming offset-, non-looming offset-, whisker-, pairing-, matched- or mismatch-responsive, based on the session in which they were significantly responsive compared to their shuffled data. From these shuffle-corrected neurons, we then used a cross-validation approach to select neurons that were positively modulated by a given stimuli on odd trials (mean response greater than zero computed from odd trials), and then plotted and analysed their responses to the even trials. This approach selected some neurons that displayed negative mean values for even trials, despite having positive mean values for their odd trials or all trials. The mismatch- and interleaved matched-responsive populations were not mutually exclusive and can show overlaps. For example, 7 neurons in Fig. 1 were classified as both mismatch and interleaved matched-responsive. Response probabilities of neurons were calculated by dividing the number of responsive trials by the total number of trials (odd and even) presented for a particular stimulus in a session. Single trial $\Delta F/F$ traces were first smoothed with a 1st-order Savitsky-Golay filter, 150 ms window. A neuron was considered responsive in a trial when its smoothed $\Delta F/F$ trace in a 1 s window from stimulus onset was significantly larger than its baseline, calculated in a 1 s window preceding the stimulus onset (comparison of 30 frames before and after the stimulus onset, Wilcoxon-Mann-Whitney test, $p < 0.05$). Population average responses presented here were determined from the averaged traces of each neuron (based on even trials) that have been identified as being positively session responsive in its respective stimulus window (selected on odd trials). The population average neuronal traces were baseline-corrected by subtracting the mean of the baseline (1 s prior to stimulus onset, corresponding to the sound presentation window) from the entire averaged trace. The mean stimulus responses were determined from single neuron average traces that were raised above zero for their minimum. For the experiments in Fig. 2 with a 1 s delay, the sound presentation window was used as the baseline. Single trial $\Delta F/F$ traces in Fig. 1b were smoothed with a 1st-order Savitsky-Golay filter, 150 ms window. No smoothing was performed for the other population average neuronal traces. We present the mean $\Delta F/F$ as a first measure of neuronal activity. We obtained comparable results for Figs. 1–4 when the selection of positively modulated neurons was based on the mean using all the trials (hence all neurons have positive means). These results are presented in Supplementary Figs. 5–8.

## Statistical analysis

All neuronal traces are presented as mean ± s.e.m. unless stated otherwise. In each box plot the central line indicates the median, the bottom and top edges of the box indicate the 25th and 75th percentiles respectively, the whiskers extend to the extreme data points (maximum and minimum points within 1.5 standard deviation), and outliers are marked with crosses. The non-parametric Wilcoxon signed-rank paired test and Wilcoxon-Mann-Whitney test for paired and unpaired group comparisons were performed respectively. All tests were two-sided. We did not test for a normal distribution of the data.

## Reporting summary

Further information on research design is available in the Nature Portfolio Reporting Summary linked to this article.

## Data availability

Source data are provided with this paper. The data used to generate the figures and reach the conclusions are available on Zenodo (https://doi.org/10.5281/zenodo.14192397). Source data are provided with this paper.

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

## Acknowledgements

We thank K. Wilmes and M. Aime for comments on an earlier version of the manuscript. We thank the members of the Nevian laboratory for comments on the manuscript. We thank C. Dellenbach, C. Käser, E. Scheuner, and J. Burkhalter for their excellent technical support in electronics and mechanics. We thank M. Känzig for his help in animal husbandry.

## Author contributions

C.R., S.K., B.L., and S.S. performed the experiments and analyzed the data. T.N. provided access to resources and materials. S.K. setup the IOS, whisker tracking, and data analysis pipeline. C.R. and S.S. designed the experiments. S.S. built the microscope and secured the funding. C.R. and S.S. wrote the manuscript with comments from all authors. This work was supported by the University of Bern and by grants to S.S. from the Swiss National Science Foundation (31003A_182678).

## Competing interests

The authors declare no competing interests.
