## [Transparent Peer Review file · Nature Communications]

Top-down modulation of sensory processing and mismatch in the mouse posterior parietal cortex

Corresponding Author: Dr Shankar Sachidhanandam

Version 0:

Reviewer comments:

Reviewer #1

(Remarks to the Author)

Raltshev et al. used a simple but cleverly designed task to search for signs of predictive processing in the mouse cortex. After teaching the mouse that a sound precedes a whisker stimulus, the authors subsequently omit the whisker stimulus on random trials and monitor how neurons in the posterior parietal cortex (PPC) and primary sensory cortex (S1) respond. Overall, I like the task, and I think this work has great potential, but the presentation could be much improved (both the results section and the figures). Specifically, there is a cacophony of neuronal response types to different stimuli under a wide variety of conditions, and I found it very hard to keep track of these - and this also prevented me from evaluating the data thoroughly. Presentation is, therefore, my main criticism. I am afraid that I don't have a suggestion for how to make the presentation more transparent and simpler, but the authors need to improve this.

Below I outline my specific criticisms and questions:

Major

* After reading the results and methods section, I was still confused about whether mice were exposed to the four trial blocks in sequence during the same session or across days. Also, it was not clear whether imaging measurements were repeated across days. Please clarify.

Line 105: "n = 6 mice, imaged across six sessions". This suggests that mice were repeatedly imaged across days, but there is no further indication later in the results.

* I was confused about the reported percentages of cells. Line 105: "3745 neurons". I understood that these are trial-modulated neurons. Then 1080 of these were significantly modulated (determined by shuffling the data). Later, the different neuronal response classes were quantified as a percentage of these 1080 neurons. Shouldn't the percentage instead be calculated in relation to the total number of cells that could be detected in a field of view? The authors need to clarify this choice.

* The argument for using either looming versus non-looming sound stimuli was unclear.

* Line 125-128: I'm not convinced this needs to be evidence of top-down mediated inhibition. There could be several untested alternative explanations. For example, could the suppression result from adaptation to repeated stimuli? Or could the suppression be explained by the mouse losing attention as the many trials go by?

* Line 145-147: Association does not necessarily happen in PPC. This could happen upstream of PPC. The association could already be in PPC's input.

* Line 174 - 181. This wasn't easy to understand.

* In the experiments introducing the delay, it is unclear when the delay was introduced. Was this a separate set of experiments?

* It was not clear what the S1 experiments brought to the table. How do these help to understand the PPC data?

* Did the authors perform the DREADD experiments (CNO injection) in mice only expressing the fluorophore and without hM4Di?

Minor

* The results of the first 1-2 figures are not unlike those that the lab presented earlier (Mohan et al. Sci. Rep), although using a technically different but conceptually similar task. This somewhat dilutes the novelty.

* Line 65. "Cingulate cortex" could be more precisely defined. Perhaps specifically, the Brodmann area?

* Why did the authors start off using RCaMP1.07, which is a less sensitive sensor than, for example, GCaMP6 or its newer variants?

* What was the field of view size, and how do the authors know that all cells are strictly in PPC, a small area that is not easy to define except with posthoc staining methods?

* Why switch Ca²⁺ indicator to study S1?

* Why is the whisker stimulus not visible in the whisker tracking plots?

* In the legends, it is repeatedly said that "The number of neurons is indicated in brackets" (for example, line 642). I searched the figures for a long time before I realized that the authors meant the numbers indicated earlier in the legend. This wasn't very clear.

Reviewer #2

(Remarks to the Author)

In this manuscript, Raltshev et al. measure mismatch responses to stimuli paired across sensory modalities in mouse PPC. This is an interesting topic, not much is known about multisensory mismatches in association regions. However, I found some fairly fundamental issues with the data analysis approach that make the data very difficult to interpret. As a result, I am not confident that they have actually observed true mismatch responses.

There are some interesting observations, such as the differences in specificity between S1 and PPC, as well as the "expectation" responses in Fig 3. Unfortunately, the central claims of the paper are not well supported by the data.

Specific concerns below:

Major:

1) Their cell selection approach is somewhat confusing and leads to the isolation of "response classes" that make up a tiny percentage of the recorded cells. For example, the "mismatch" neurons represent only 68 out of 3745 neurons (~1.8%). Although I had trouble figuring out exactly what their noise floor is, such a small number of neurons may be due purely to chance. Even of that small subset, a number of the neurons do not respond robustly (Fig. 1c). This means that the "mismatch" response class, which much of their later analysis is based on, may well be a statistical artifact.

2) Related to the first point, the response types are sorted into experimenter-defined categories based on their max response during a particular task epoch. As a result, averaging across sorted neurons could lead to an artificial "response" resulting even from random noise. It is important to use a cross-validated approach here. For example, the neurons could be sorted into response classes based on odd trials, and their responses plotted using even trials.

3) Finally, my greatest concern is that many of the supposed "mismatch" neurons begin firing *prior* to the omitted stimulus. This can be seen clearly in Fig 1d, where the responses rise partway through the auditory stimulus for the mismatch but not the paired trials. This does not make any sense as a mismatch response, as during the auditory stimulus, the neuron does not yet know whether the whisker stimulus will be omitted or not. This "acausal" response to mismatches is present in several of the other analyses of mismatch responses (Fig 3c, 3f, and Fig S4b). This is especially obvious in Fig 3c, where the "mismatch" response begins several seconds before the eventual omitted stimulus. I suspect this puzzling result is due to the artificial sorting of neural responses into response types described in points 1 and 2.

Minor:

1) Reference 6 (Saleem et al) actually argued against a predictive coding interpretation of their data. They argued that visual motion and speed were independently integrated.

2) The localization of area A is not very clear from the intrinsic signal imaging. Given the small size of the region, is there a better way to localize it to make sure the same area is being targeted across mice?

3) I found the reporting of the percentage of responsive cells confusing. For example, in line 116, it says that 27% of PPC neurons were tuned to whisker deflection, but the denominator only includes responsive cells and not all PPC neurons (actual value is 7.8%). Also, I was not sure what they meant by response probability? (I might have missed it, but I don't believe it was defined anywhere).

4) Line 236: It is unlikely that CNO administration will "silence" the M2 feedback projections. At best, the DREADD activation

might reduce activity, though this is only putative since there was no data shown. Also, there were no CNO-only controls to rule out off-target effects of the drug.

5) "Thy-1-GCaMP6f" should be "Thy1-GCaMP6f" throughout.

Reviewer #3

(Remarks to the Author)

In this study, Raltshev and colleagues exposed mice to auditory-to-tactile stimulus sequences and investigated predictive and prediction error signals in PPC. Previous work in mice has proposed that cortex possesses two types of prediction error neurons which signal either stronger or weaker sensory input than predicted. Here they report that the mouse PPC possesses similar types of prediction error neurons and conclude that the secondary motor cortex is the source of predictive inputs. Overall, this is a very interesting approach to identifying potential comparator circuits in PPC that might employ a predictive processing algorithm, but we fear, currently the evidence in the manuscript is not quite strong enough to fully support their arguments. Assuming the authors can address these concerns, we would recommend publication.

This review was written in collaboration with a postdoc in the lab.

Major

1. We fear there appears to be a potentially serious and central problem with the manuscript currently. The whisker stimulus omission responses all appear to occur well before the actual omission of the whisker stimulus. E.g. in Figure 1d, the bottom row, rightmost panel (omission responses of "interleaved mismatch" neurons), the response starts rising during the looming sound (i.e. before the omission). The actual omission does not appear to trigger any response. The same is apparent in the Figure 3d. So, this does not appear to be an omission response – and it can't really be an expectation response either, as the response is absent in "interleaved paired" trials (the mouse cannot know whether the stimulus will be omitted during the looming sound yet (we assume given the description of the task). We see no possibility for the responses – with the timing as currently shown – to be real effects (as opposed to some analysis error).

The most frequent explanation we have encountered for responses that look like this is an analysis error related to a regression to the mean effect. The error comes from selecting a subset of neurons responsive to stimulus X, and then plotting the response of these neurons to stimulus X. If selection of the neurons and plotting of the response is performed on the same data, this results in circular analysis https://en.wikipedia.org/wiki/Circular_analysis (a variant of a regression to the mean effect). One way to prevent this is to use half the data (e.g. odd trials) to select neurons responsive for stimulus X, and then use the other half of the data (e.g. even trials) to plot responses of these neurons to stimulus X. This prevents the selection from interfering with response. If the authors are already performing the analysis in this way and we missed it, the only other explanation we can think of is that the stimulus onset times in Figures 1 and 3 are systematically shifted? Or the authors chose some smoothing of the data that artificially shifts responses?

Minor

2. Please do not separate figures and figure legends – this is a practice that dates from a time before the advent of computers! And makes reviewing irritating. Related to this, please use a formatting of references that is readable without going to the bibliography (i.e. just use the (Name et al, year) format). You are not formatting for the journal publication yet.

3. It is unclear when mice formed the predictions. In lines 125 - 128, the authors state that mice have predictions already in pairing sessions based on the reduced auditory responses relative to sound sessions (Figure 1d and 1e). In contrast, whisker responses were not reduced in pairing sessions compared with whisker sessions (Figure 1e) but were reduced if interleaved paired sessions were compared with pairing sessions. Also, ramping activity of interleaved paired neurons was only observed in interleaved paired sessions but not in pairing sessions (Figure 2f) (even more confusingly, there seems ramping activity in pairing sessions in Figure 4g). These observations suggest that predictions were not yet formed in pairing sessions. The authors should make their view much clearer. If they consider that predictions form during pairing sessions, fine descriptions of how pairing sound responses emerge, and sound looming responses disappear would be helpful.

4. Figures 2d and 2i are mildly confusing in presentation to put it mildly. "pairing" in the panel title, "paired" and "pairing" on the x-axis labels? Please fix.

5. It is unclear at which time scales different sessions (e.g. "sound" session, "whisker" session, and etc) were introduced in this paradigm. Are they within a single day? Without having any breaks between sessions?

6. line 128: be consistent either learned or learnt.

7. line 207: Fig. 3a should probably be Fig. 3e.

8. Locations of cranial windows should probably differ between Figures 1 and 2

9. What is the whisker trace in Figure 1d? A trial average?

10. Figure 3a and 3c: please add legends.

11. Figure 3c,f: showing the orange ticks in mismatch (omission) trials is confusing.
12. Figure 4a: what does the fluorescent image show?
13. The authors often failed to refer figures that support their statements in the manuscript (e.g. line 115). Also, it is often hard to find which figure panels the readers should compare (e.g. lines 123 - 125).
14. Ref 19 does not talk about “hierarchical visual processing” – possibly the wrong reference?
15. It is difficult to assess whether the imaged region actually corresponds to area A of PPC. The authors just located S1 and V1 using intrinsic imaging and defined the rest as PPC. Assuming the authors did not further pinpoint area A specifically, we would recommend just referring to the target area as PPC, not area A. Area A is relatively small and hard to hit without detailed mapping.
16. Why does sound offset response start before the offset (Figure 1d) even using non-rooming sounds (Figure S2 b). It is probably incorrect to refer to these neurons as offset-responsive. See major comment 1 – unclear what is going on. Similarly, the neurons that are defined as whisker responsive start responding before the tactile onsets (Figure 1d, row 3 column 2).
17. The number of S1 neurons that are responsive both in whisker and pairing sessions is surprisingly small (Figure 2c). This makes us suspect that substantial part of the response changes in pairing sessions could come from sensory adaptation.
18. What is compared in Figure 3g?
19. The DREADD experiments should be ideally performed with control experiment (e.g. CNO+ w/o receptor expression).
20. Line 251 - 253: is this significant?
21. The DREADD analysis in Figure 5 should probably be done on actual population averages (i.e. the entire population). First selecting responsive neurons and then plotting the responses of these subsets has the confound that the fraction of neurons in these subsets will change? May we are misunderstanding something here.
22. The argument in Figure S5 is hard to understand. The authors’ argument reads like the introduction of mismatch increases top-down predictive inputs. Or did they want to say prediction was not formed completely during the pairing sessions?
23. Are the serotypes and pseudotypes of the AAVs correct? We suspect the AAVs used were AAV2/1 (not AAV1/2 or AAV2).

Version 1:

Reviewer comments:

Reviewer #1

(Remarks to the Author)

The authors have effectively addressed the points I raised. Congratulations on producing an engaging and thought-provoking paper.

Reviewer #2

(Remarks to the Author)

After reading the revised submission, I am sorry to say that the manuscript still suffers from many of the same statistical errors and circular reasoning present in the original submission.

1) In regards to major point 1 in the original critique: The authors report that “3.7% of neurons (72 of 1923) can report the whisker stimulus omission in Fig.1”. Given that the alpha value for their Mann-Whitney test is 0.05, we would expect ~5% of neurons to have significant responses purely by chance, which is greater than the reported number of mismatch responses.

In the rebuttal letter, they then start adding the results of different tests to claim that a larger percentage of the neurons respond to mismatch. However, if each statistical test has a 5% false positive rate, then we would expect running two statistical tests (e.g., for 0 ms delay and 1000 ms delay) and adding up the “significant” cells would yield ~10% responsive cells purely by chance, similar to their measured value of 12.9%. I remain unconvinced that the “mismatch” responses (that

all subsequent analyses rely on) are not just a statistical artifact.

2) In regards to major point 2 in the original critique (also pointed out by reviewer #3): Their reanalysis did not satisfactorily address the circular analysis issue. They continue to average responses from neurons that were determined a priori to be responsive during a particular epoch without cross-validation. This is clear in their figures, where a sharp band of elevated activity can be seen running diagonally through the responses (Fig 1a, Fig 3a, c). As reviewer #3 pointed out, this is a classic example of circular analysis.

Later in the response letter, the authors state that they tried cross-validation, but that it led to slightly noisier responses (showing a single non-mismatch cell as an example). However, the slightly higher noise level in the responses is much preferable to plotting the responses of neurons preselected for responding during a particular interval. Plotting cross-validated responses is now standard procedure in the field.

3) In regards to major point 3 in the original critique (again pointed out by reviewer #3): The authors state that they fixed an error in the code that eliminated the acausal responses observed in the original manuscript. This mostly appears to be the case, but there do appear to still be “mismatch” responses occurring before the absent stimulus in Fig 5c.

Reviewer #3

(Remarks to the Author)

The authors have done a commendable job addressing most of the concerns. There is one remaining concern that is still a bit worrying (major comment 2) – but I assume we are just misunderstanding something here. If this is indeed the case and the authors have a good explanation, we would recommend publication.

This review was written in collaboration with a postdoc in the lab.

Major

1. Remove Figure 2. This adds very little in its current form (following Figure 1, what one would really want to know is if there are mismatch responses in S1, and they are conspicuously absent). More troubling, is that given that the authors likely have not regression to the mean corrected the analysis here (given that they chose not to do it elsewhere “because it looks too noisy”...), this is almost certainly all a regression to the mean effect. If you select the neurons that are most responsive to stimulus A, these neurons will automatically look like they are less responsive to stimulus B and C, simply as a function of regression to the mean. How big this effect is depends on the trial-to-trial variability of the neurons. Thus, all this figure shows is that the trial-to-trial noise is likely higher in PPC than it is in S1. But most importantly, without a comparison of the mismatch responses between S1 and PPC it adds very little to the manuscript. Just remove it.

2. Line 122. 3.7% of neurons were mismatch responsive. We don't understand, if $p < 0.05$ was used to define “responsive”, any % of responsive neurons of 5% or less would mean the fraction of responsive neurons is at – or below – what one would expect to find by chance. I.e. 3.7% of mismatch responsive neurons would translate to finding “no evidence for mismatch responsive neurons”. I assume we are misunderstanding something here.

Minor

3. The usage of “paired” and “pairing” in the figure legends (Figure 1e, Figure 2b,d,f,h, etc.) is still confusing. On top of this the authors use these terms to talk about neuron types and session types in the same figure (e.g. Fig 2 a, d, e & h). We would suggest to remedy this.

Version 2:

Reviewer comments:

Reviewer #2

(Remarks to the Author)

The second revision of their manuscript makes a few improvements on the previous version, but the authors appear to insist on using incorrect approaches for the analysis and display of their data.

1) In the last review, both reviewer #3 and I pointed out that there were fewer significant “prediction error” responsive cells (3.7%) than would be predicted by chance given the 0.05 alpha value used for their statistical tests. The authors found a mistake in their code that, once corrected, raised the number of significant cells to 8.6%. This is still a relatively small fraction of cells (~3.6% above that expected purely by chance), so there is some question of how prevalent these signals are in the population, but at least it is above the level expected by chance.

I'm somewhat troubled that this is the second major mistake in their code that they've discovered during the review process (the other one caused “acausal” prediction error responses). Both of the errors had substantial effects on the direction and magnitude of the reported findings. Although I don't have the time to review the code carefully, I am concerned by rigor of their analysis code if these major errors are still being uncovered (and likely would not have been discovered if they didn't directly relate to reviewer critiques).

2) For reasons I fail to grasp, the authors are very resistant to using cross-validated responses, though this is clearly the correct way to analyze/plot the data (as pointed out by multiple reviewers). They showed us a few more example cells in the response to reviewers document, but continue to use the “circular” approach for displaying the data in the paper. For all analyses using “pre-selected” cells, the authors need to plot/analyze the cross-validated responses. This should be done throughout the paper, including in later Figures using similar selection criteria (e.g., Figure 4a-b).

3) I am still confused by the findings of Fig 4. It looks like the entire population response is reduced in the M2>PPC inhibition experiments, not specifically the mismatch response. Is it really accurate to say that “M2 contributes to the top-down prediction to the PPC”, specifically?

Reviewer #3

(Remarks to the Author)

A few minor remaining points:

1. The authors should really show cross-validated plots (as they are shown in response to review 2) instead of what they show in the paper. E.g. the very bizarre looking response to the 1s delay mismatch is not there anymore when using the proper cross validated way of plotting.
2. I am not sure if the figure is new or different from previous submissions (I may have also missed in the previous review), but the difference in Figure 4c is likely not driven by the mismatch response? Primarily the sound response appears to be affected by M2 silencing.
3. I have pointed this out before, but the “paired” “pairing” terminology does not work. Figure labels of the form “post-stimulus interleaved paired – pairing” (as e.g. in Figure 3c – and shouldn’t panels c and e have different labels) are not intelligible.

Version 3:

Reviewer comments:

Reviewer #2

(Remarks to the Author)

This is the third revision of this paper I have reviewed. Although the authors eventually addressed the more serious statistical and data visualization errors raised by the reviewers, my view throughout has been that the findings are not particularly noteworthy. Specifically, the mismatch cells the paper is based on are both few in number and exhibit weak responses, barely distinguishable from noise. Moreover, despite strong claims that PPC-projecting M2 neurons are responsible for the mismatch responses, I found the chemogenetic results to be unconvincing, with no effect on the mismatch neurons themselves, and somewhat uninterpretable effects on the population averages. For these reasons, I do not believe the manuscript warrants publication in Nature Communications.

However, it appears that I am in the minority, and I do not see value in reviewing further revisions of this manuscript. If the manuscript is published, I ask that the issues I raised in the review process are included in the reviewer report so the readers can see the reasons for my dissent.

REVIEWER COMMENTS

Reviewer #1 (Remarks to the Author):

Raltshev et al. used a simple but cleverly designed task to search for signs of predictive processing in the mouse cortex. After teaching the mouse that a sound precedes a whisker stimulus, the authors subsequently omit the whisker stimulus on random trials and monitor how neurons in the posterior parietal cortex (PPC) and primary sensory cortex (S1) respond. Overall, I like the task, and I think this work has great potential, but the presentation could be much improved (both the results section and the figures). Specifically, there is a cacophony of neuronal response types to different stimuli under a wide variety of conditions, and I found it very hard to keep track of these - and this also prevented me from evaluating the data thoroughly. Presentation is, therefore, my main criticism. I am afraid that I don't have a suggestion for how to make the presentation more transparent and simpler, but the authors need to improve this.

We thank the reviewer for the critical comments that have helped to improve the manuscript. As the manuscript is centered around whisker stimulus evoked responses and subsequent mismatches thereof, we have moved the sound evoked data to the supplementary figures. We now focus on and describe the responses of neurons in the whisker, pairing and subsequent interleaved sessions (with paired and mismatch trials). We believe that this greatly simplifies the presentation of the data, along with its interpretation.

Below I outline my specific criticisms and questions:

Major

** After reading the results and methods section, I was still confused about whether mice were exposed to the four trial blocks in sequence during the same session or across days. Also, it was not clear whether imaging measurements were repeated across days. Please clarify.*

Line 105: "n = 6 mice, imaged across six sessions". This suggests that mice were repeatedly imaged across days, but there is no further indication later in the results.

We apologize for not describing the experimental sequence in a clear enough manner. Mice were exposed to the four trial blocks (sessions in sequence during the same imaging session under the microscope. Some mice were imaged across days but this was performed in different field of views. The data in Fig1 was acquired from a single field of view for each of the 6 mice, and we clarify this in the manuscript line106. For the data in Fig 2, it was acquired from 4 mice, from 6 and 8 fields of view, for S1 and PPC respectively, over different sessions performed 1 to 2 weeks apart. We now clarify this in lines 151 and 158.

** I was confused about the reported percentages of cells. Line 105: "3745 neurons". I understood that these are trial-modulated neurons. Then 1080 of these were significantly modulated (determined by shuffling the data). Later, the different neuronal response classes were quantified as a percentage of these 1080 neurons. Shouldn't the percentage instead be calculated in relation to the total number of cells that could be detected in a field of view? The authors need to clarify this choice.*

We agree with the reviewer and now quantify the number of session responsive neurons (whisker, pairing, paired and mismatch) as a percentage of the total number of neurons detected in the corresponding field of view. We had an error in our analysis code for counting the detected cells per field of view and have now updated the numbers in the manuscript. We apologize for the confusion caused by the previous quantification.

** The argument for using either looming versus non-looming sound stimuli was unclear.*

We previously showed that mismatch responses to the absence of a tactile stimulus cued by a looming sound could be generated in the PPC. As such we used the looming sound in the auditory tactile sensory sequence to predict the whisker stimulus. Nonetheless in a separate set of control experiments (See Supplementary Fig. 2) we demonstrate that mismatch responses can also be generated with a non-looming sound as part of the sensory sequence. We now clarify the choice of looming versus non-looming sound in the Methods.

** Line 125-128: I'm not convinced this needs to be evidence of top-down mediated inhibition. There could be several untested alternative explanations. For example, could the suppression result from adaptation to repeated stimuli? Or could the suppression be explained by the mouse losing attention as the many trials go by?*

We agree with the reviewer that the suppression could be due to several untested alternative explanations. As we no longer focus on sound evoked responses in the manuscript, we have removed this statement.

** Line 145-147: Association does not necessarily happen in PPC. This could happen upstream of PPC. The association could already be in PPC's input.*

We agree with the reviewer that sensory association does not necessarily need to originate in the PPC and have adapted the sentence accordingly.

** Line 174 - 181. This wasn't easy to understand.*

Here we wanted to point out that the interleaved paired-responsive neurons in the PPC were the only ones that showed an increased response to the whisker stimulus, between the pairing and interleaved sessions. All other neurons displayed a stable or decreased whisker stimulus response. These interleaved paired-responsive neurons in the PPC further displayed an increase in the pre-stimulus response, when comparing between the interleaved and pairing session. Despite these increases in neuronal activity, their response to the whisker stimulus was smaller than that of pairing-responsive neurons in the pairing session. We now clarify this in the text.

** In the experiments introducing the delay, it is unclear when the delay was introduced. Was this a separate set of experiments?*

Yes, this was done in a separate set of experiments. Here we performed two sets of interleaved sessions (approximately 120 paired trials with 30 randomly interleaved mismatch trials), one after the other within the same field of view. The first set had no delay between the sound and whisker stimulus and the delay of 1s was introduced in the second set. We now clarify this in lines 179 to 184 in the manuscript.

** It was not clear what the S1 experiments brought to the table. How do these help to understand the PPC data?*

The S1 data was essential to illustrate the dynamic nature of sensory processing in the PPC. There was more overlap in the responsiveness of neurons to the whisker stimulus between the sessions in S1, indicating that these S1 neurons could reliably represent the whisker stimulus. In contrast in the PPC, apart from the interleaved paired-responsive neurons, neurons tend to respond to the whisker stimulus only in their respective session. Hence the PPC can also reliably represent the whisker stimulus in each session, albeit with a different composition of neurons.

** Did the authors perform the DREADD experiments (CNO injection) in mice only expressing the fluorophore and without hM4Di?*

This is a good suggestion and we have now included these control experiments in our revised manuscript in Fig.4 and 5. We also describe the effects of CNO injection on sensory processing and mismatch in Supplementary Fig. 5.

Minor

** The results of the first 1-2 figures are not unlike those that the lab presented earlier (Mohan et al. Sci. Rep), although using a technically different but conceptually similar task. This somewhat dilutes the novelty.*

While we agree that our experiments are conceptually similar and inspired by our earlier work, the results in Fig. 1 and Fig. 2 illustrate for the first time (to our knowledge) the highly dynamic nature of sensory processing in the PPC, in comparison to S1. These findings would not have been possible in Mohan et al, as it was impossible to separate the presentation of the auditory and tactile stimuli, which were an integral part of the texture presentation apparatus. Hence on the contrary we believe that the findings in Fig.1 and Fig.2 are novel, compared to Mohan et al.

** Line 65. "Cingulate cortex" could be more precisely defined. Perhaps specifically, the Brodmann area?*

We have now replaced cingulate cortex with anterior cingulate cortex A24 and midcingulate cortex A24' in line 64.

** Why did the authors start off using RCaMP1.07, which is a less sensitive sensor than, for example, GCaMP6 or its newer variants?*

When we started these experiments in the PPC, our custom built 2-photon microscope was equipped with a single wavelength fiber laser at 1064nm, which limited us to the use of the red shifted GECI RCaMP1.07. A substantial part of the experiments were done with RCaMP1.07 before we had access to a tunable laser which allowed us to image with GCaMP6. The use of 2 different GECIs allowed us to further cross-validate the dynamic nature of sensory processing in the PPC, in response to audio-tactile sensory sequences.

** What was the field of view size, and how do the authors know that all cells are strictly in PPC, a small area that is not easy to define except with posthoc staining methods?*

The viral injection for GECI expression in the PPC is first done at 1.7mm lateral and 2.0mm posterior of bregma, and the cranial window is placed over this site. Before the start of imaging experiments, we perform intrinsic optical signal (IOS) imaging to determine the boundaries of V1 and the γ -barrel-column of S1 at the site of the cranial window (described in Methods lines 367 to 374). We have previously shown that the area between these boundaries corresponds to approximately area A of the PPC, using anatomical tracing methods (Mohan et al, 2018). We have a field of view size that is approximately 450 by 450 microns. By using the blood vessel patterns identified from the IOS as map, we restrict our field of view largely to area A of the PPC. It is well possible that we might also image the boundaries of areas RL and AM of the PPC. At the end of the experiments mice are perfused and the brains recovered and sliced to confirm the location of the PPC, now stated in line 451.

** Why switch Ca²⁺ indicator to study S1?*

We wanted to compare the neuronal responses in S1 and PPC to the audio-tactile sequence with the same GECI. As such we used the Thy1-GCaMP6f in these experiments as we were able to image S1 and PPC in the same mice with a single cranial window in place.

** Why is the whisker stimulus not visible in the whisker tracking plots?*

The whisker deflection effectuated by the magnetic field is in the vertical axis and is subsequently not captured by the whisker tracking camera which tracks lateral whisker movements. We have previously extensively quantified the whisker deflections delivered with the magnetic coil, using a laser displacement sensor (Sachidhanandam et al, 2013, Nat. Neurosci.).

** In the legends, it is repeatedly said that "The number of neurons is indicated in brackets" (for example, line 642). I searched the figures for a long time before I realized that the authors meant the numbers indicated earlier in the legend. This wasn't very clear.*

We now state the number of neurons as x neurons, and apologize for the lack of clarity in the legends.

Reviewer #2 (Remarks to the Author):

In this manuscript, Raltshev et al. measure mismatch responses to stimuli paired across sensory modalities in mouse PPC. This is an interesting topic, not much is known about multisensory mismatches in association regions. However, I found some fairly fundamental issues with the data analysis approach that make the data very difficult to interpret. As a result, I am not confident that they have actually observed true mismatch responses.

There are some interesting observations, such as the differences in specificity between S1 and PPC, as well as the "expectation" responses in Fig 3. Unfortunately, the central claims of the paper are not well supported by the data.

We thank the reviewer for the critical comments that have helped to improve the manuscript. We have completely re-analyzed our data as suggested and we are now confident that we observe true mismatch responses with respect to the timing of the predicted whisker stimulus, as we describe below.

Specific concerns below:

Major:

1) Their cell selection approach is somewhat confusing and leads to the isolation of "response classes" that make up a tiny percentage of the recorded cells. For example, the "mismatch" neurons represent only 68 out of 3745 neurons (~1.8%). Although I had trouble figuring out exactly what their noise floor is, such a small number of neurons may be due purely to chance. Even of that small subset, a number of the neurons do not respond robustly (Fig. 1c). This means that the "mismatch" response class, which much of their later analysis is based on, may well be a statistical artifact.

In our revised manuscript, we now report that 3.7% of neurons (72 of 1923) can report the whisker stimulus omission in Fig.1. These neurons have a response probability of 34.7% to the stimulus omission trials, which is comparable to that of the interleaved paired-responsive neurons (545 of 1923, 28.3%) which have a response probability of 30.9%. In Fig. 3c, we report that 5.9% (79 of 1335) and 7% (94 of 1335) of neurons can report the mismatch trials at 0ms and 1000ms delay respectively, with an overlap of 1 neuron. Hence here mismatch neurons make up 12.9% (172 of 1335) of all neurons. In Fig. 3g we report that 7.6% (124 of 1631) and 8.6% (140 of 1631) of neurons can report omission and decreased intensity mismatch trials respectively, while 7.1% (116 of 1631) of neurons can report increased

intensity mismatch trials. Here mismatch neurons would make up 26.7% (364 of 1361) of all neurons. Hence approximately 7% of PPC neurons can report a given mismatch type, with largely different neurons reporting these mismatches. We are thus convinced that mismatch neurons in the PPC are not a statistical artifact.

2) Related to the first point, the response types are sorted into experimenter-defined categories based on their max response during a particular task epoch. As a result, averaging across sorted neurons could lead to an artificial “response” resulting even from random noise. It is important to use a cross-validated approach here. For example, the neurons could be sorted into response classes based on odd trials, and their responses plotted using even trials.

In order to have an unbiased approach to the selection of stimulus responsive neurons, we reanalyzed our data in the following manner: For each detected neuron, we present the raw fluorescence traces (F) of the calcium signals as $\Delta F/F = (F - F_0)/F_0$, where F_0 was calculated for each trial as the mean of the 1 s window prior to the sound cue that signalled trial start. We did not perform any denoising using deconvolution. The responsiveness of a detected neuron to a given stimulus (sound, sound-offset, whisker, mismatch) was determined by comparing the distribution of its single trial responses (single trial mean $\Delta F/F$ calculated in a 1 s window from stimulus onset minus the mean baseline window 1 s before stimulus onset) against the distribution of 1000 randomly selected events from its same session (random baseline corrected mean $\Delta F/F$ calculated in a 1 s window as above), hence taking into account the random noise of each neuron. Significance was determined with a two-sided Mann-Whitney-U test ($p < 0.05$). We thus identified populations of neurons that were positively or negatively modulated by a given stimuli. We subsequently classified the positively modulated shuffle-corrected neurons (determined by the mean stimulus response being positive on average) as looming-, non-looming-, looming offset-, non-looming offset-, whisker-, pairing-, paired- or mismatch-responsive, based on the session in which they were significantly responsive compared to their shuffled data. The mismatch- and interleaved paired-responsive populations were not mutually exclusive. We have now updated the analysis section in the revised manuscript accordingly, in lines 459 to 468.

*3) Finally, my greatest concern is that many of the supposed “mismatch” neurons begin firing *prior* to the omitted stimulus. This can be seen clearly in Fig 1d, where the responses rise partway through the auditory stimulus for the mismatch but not the paired trials. This does not make any sense as a mismatch response, as during the auditory stimulus, the neuron does not yet know whether the whisker stimulus will be omitted or not. This “acausal” response to mismatches is present in several of the other analyses of mismatch responses (Fig 3c, 3f, and Fig S4b). This is especially obvious in Fig 3c, where the “mismatch” response begins several seconds before the eventual omitted stimulus. I suspect this puzzling result is due to the artificial sorting of neural responses into response types described in points 1 and 2.*

In our reanalyzed data, the mismatch responses no longer begin prior to the omitted stimulus, as we now show in Fig. 1d, Fig 3c, 3f and Supplementary Fig 3b (formally Fig S4b). We believe that this was an analysis artifact as a result of inaccurate denoising on our part in the original manuscript and we apologize for this oversight.

As suggested, we tried a cross-validated approach to select for positively modulated stimulus responsive neurons by using the odd trials to determine the mean stimulus response and plotting the even trials. Below we show the population averages of $\Delta F/F$ traces of interleaved

paired-responsive neurons with a 1000 ms delay, selected on odd trials and all trials. While both averages look comparable in terms of response onset, the selection on odd trials is more noisy as only half the trials are used in plotting. As such in the revised manuscript we selected for responsive neurons using all the trials.

Figure for reviewer

Minor:

1) Reference 6 (Saleem et al) actually argued against a predictive coding interpretation of their data. They argued that visual motion and speed were independently integrated.

We apologize for this oversight on our part and have replaced the reference accordingly.

2) The localization of area A is not very clear from the intrinsic signal imaging. Given the small size of the region, is there a better way to localize it to make sure the same area is being targeted across mice?

We agree with the reviewer that area A is a small region making it potentially difficult to localize. Nonetheless we are able to localize it systematically across mice by using intrinsic optical signal imaging to localize the bordering γ -barrel-column of S1 and the boundaries of V1. Unfortunately, we know of no better or simpler way to localize it to facilitate targeting across mice.

3) I found the reporting of the percentage of responsive cells confusing. For example, in line 116, it says that 27% of PPC neurons were tuned to whisker deflection, but the denominator only includes responsive cells and not all PPC neurons (actual value is 7.8%). Also, I was not sure what they meant by response probability? (I might have missed it, but I don't believe it was defined anywhere).

We now quantify the number of session responsive neurons (whisker, pairing, paired and mismatch) as a percentage of the total number of neurons detected in the corresponding field of view. We had an error in our analysis code for counting the detected cells per field of view and have now updated the numbers in the manuscript. We apologize for the confusion caused by the previous quantification.

The response probabilities of probabilities of neurons were calculated by dividing the number of responsive trials by the total number of trials presented for a particular stimulus in a session. Single trial $\Delta F/F$ traces were first smoothed with a 1st-order Savitsky-Golay filter, 150ms window. A neuron was considered responsive in a trial when its smoothed $\Delta F/F$ trace in a 1 s window from stimulus onset was significantly larger than its baseline, calculated in a 1 s window preceding the stimulus onset different (30 frames before and after the stimulus

onset, Wilcoxon-Mann-Whitney test). We have now updated this definition in the analysis section, in lines 475 to 482.

4) Line 236: *It is unlikely that CNO administration will “silence” the M2 feedback projections. At best, the DREADD activation might reduce activity, though this is only putative since there was no data shown. Also, there were no CNO-only controls to rule out off-target effects of the drug.*

We agree with the reviewer that CNO only control experiments are essential in order to better interpret the influence of inhibitory DREADD activation on M2 feedback projections. As such we have performed these experiments and describe our findings in Fig.4 and Fig.5 of the revised manuscript.

5) *“Thy-1GCaMP6f” should be “Thy1-GCaMP6f” throughout.*

We thank the reviewer for this correction, and we have updated the manuscript accordingly.

Reviewer #3 (Remarks to the Author):

In this study, Raltshev and colleagues exposed mice to auditory-to-tactile stimulus sequences and investigated predictive and prediction error signals in PPC. Previous work in mice has proposed that cortex possesses two types of prediction error neurons which signal either stronger or weaker sensory input than predicted. Here they report that the mouse PPC possesses similar types of prediction error neurons and conclude that the secondary motor cortex is the source of predictive inputs. Overall, this is a very interesting approach to identifying potential comparator circuits in PPC that might employ a predictive processing algorithm, but we fear, currently the evidence in the manuscript is not quite strong enough to fully support their arguments. Assuming the authors can address these concerns, we would recommend publication.

We thank the reviewer for the critical comments that have helped to improve the manuscript. We have completely re-analyzed our data as suggested and we are now able to address the raised concerns as described below.

This review was written in collaboration with a postdoc in the lab.

Major

1. *We fear there appears to be a potentially serious and central problem with the manuscript currently. The whisker stimulus omission responses all appear to occur well before the actual omission of the whisker stimulus. E.g. in Figure 1d, the bottom row, rightmost panel (omission responses of “interleaved mismatch” neurons), the response starts rising during the looming sound (i.e. before the omission). The actual omission does not appear to trigger*

any response. The same is apparent in the Figure 3d. So, this does not appear to be an omission response – and it can't really be an expectation response either, as the response is absent in “interleaved paired” trials (the mouse cannot know whether the stimulus will be omitted during the looming sound yet (we assume given the description of the task). We see no possibility for the responses – with the timing as currently shown – to be real effects (as opposed to some analysis error).

The most frequent explanation we have encountered for responses that look like this is an analysis error related to a regression to the mean effect. The error comes from selecting a subset of neurons responsive to stimulus X, and then plotting the response of these neurons to stimulus X. If selection of the neurons and plotting of the response is performed on the same data, this results in circular analysis https://en.wikipedia.org/wiki/Circular_analysis (a variant of a regression to the mean effect). One way to prevent this is to use half the data (e.g. odd trials) to select neurons responsive for stimulus X, and then use the other half of the data (e.g. even trials) to plot responses of these neurons to stimulus X. This prevents the selection from interfering with response. If the authors are already performing the analysis in this way and we missed it, the only other explanation we can think of is that the stimulus onset times in Figures 1 and 3 are systematically shifted? Or the authors chose some smoothing of the data that artificially shifts responses?

We have completely reanalyzed our data as we have described in our response to point 2 of Reviewer 2. In our revised manuscript, the mismatch responses no longer begin prior to the omitted stimulus, as we now show in Fig. 1d, Fig 3c, 3f and Supplementary Fig 3b (formally Fig S4b). We believe that this was an analysis artifact as a result of inaccurate denoising on our part in the original manuscript and we apologize for this error. We now present $\Delta F/F$ traces determined from the raw fluorescence and not denoised traces.

As suggested, we tried a cross-validated approach to select for positively modulated stimulus responsive neurons by using the odd trials to determine the mean stimulus response and plotting the even trials. In our response to point 3 of Reviewer 2 we show the population averages of $\Delta F/F$ traces of interleaved paired-responsive neurons with a 1000 ms delay, selected on odd trials and all trials respectively. While both averages look comparable in terms of response onset, the selection on odd trials is more noisy as only half the trials are used in plotting. As such in the revised manuscript we selected for responsive neurons using all the trials.

Minor

2. Please do not separate figures and figure legends – this is a practice that dates from a time before the advent of computers! And makes reviewing irritating. Related to this, please use a formatting of references that is readable without going to the bibliography (i.e. just use the (Name et al, year) format). You are not formatting for the journal publication yet.

We apologize for the inconvenience in the reading the manuscript caused by our choice of formatting. We have rectified this accordingly in the revised manuscript. Unfortunately for the references we are now obliged to use the numbering format for the revised manuscript and apologize once again for the inconvenience.

3. It is unclear when mice formed the predictions. In lines 125 - 128, the authors state that mice have predictions already in pairing sessions based on the reduced auditory responses relative to sound sessions (Figure 1d and 1e). In contrast, whisker responses were not reduced in pairing sessions compared with whisker sessions (Figure 1e) but were reduced if interleaved paired sessions were compared with pairing sessions. Also, ramping activity of interleaved paired neurons was only observed in interleaved paired sessions but not in pairing sessions (Figure 2f) (even more confusingly, there seems ramping activity in pairing sessions in Figure 4g). These observations suggest that predictions were not yet formed in pairing sessions. The authors should make their view much clearer. If they consider that predictions form during pairing sessions, fine descriptions of how pairing sound responses emerge, and sound looming responses disappear would be helpful.

We agree with the reviewers that it is unlikely that predictions were formed during the pairing sessions. We no longer observe the early ramping /enhanced pre-stimulus activity in the pairing sessions, as we now show in Fig.1d, Fig. 2e and Supplementary Fig. 4d (formally Fig. 4g). As such we updated the revised manuscript accordingly.

4. Figures 2d and 2i are mildly confusing in presentation to put it mildly. "pairing" in the panel title, "paired" and "pairing" on the x-axis labels? Please fix.

We apologize for the confusing labels, and we have now rectified this in the updated Fig. 2d and 2h.

5. It is unclear at which time scales different sessions (e.g. "sound" session, "whisker" session, and etc) were introduced in this paradigm. Are they within a single day? Without having any breaks between sessions?

We now state that the sessions were presently sequentially within the same imaging session, in line 96-97. We apologize for the lack of clarity in the original manuscript.

6. line 128: be consistent either learned or learnt.

We have adapted the revised manuscript accordingly.

7. line 207: Fig. 3a should probably be Fig. 3e.

We thank the reviewer for pointing out this error and we have rectified this.

8. Locations of cranial windows should probably differ between Figures 1 and 2

We use 4mm cranial windows that are approximately centered over the PPC at 1.7mm lateral and 2.0mm posterior of bregma. Hence we are able to image neurons in column 1 of S1, just anterior to the γ barrel that we locate using intrinsic optical signal imaging, as well as the PPC.

9. What is the whisker trace in Figure 1d? A trial average?

The whisker traces are trial averages that we acquired simultaneously with the single neuron examples.

10. Figure 3a and 3c: please add legends.

We have now included legends for these figures.

11. Figure 3c,f: showing the orange ticks in mismatch (omission) trials is confusing.

We have included a legend that in these figures that indicates the presence and absence/change in intensity of the whisker stimulus for the interleaved paired and mismatch trials respectively. We believe that showing the location of the whisker stimulus is important for the paired trials of the mismatch neurons and the inclusion of the legend should help to reduce confusion.

12. Figure 4a: what does the fluorescent image show?

This image shows the expression of mCitrine to show the presence and location of M2 neurons that express the inhibitory DREADD and project to the PPC.

13. The authors often failed to refer figures that support their statements in the manuscript (e.g. line 115). Also, it is often hard to find which figure panels the readers should compare (e.g. lines 123 - 125).

We now refer to the corresponding figures and panels to support our statements.

14. Ref 19 does not talk about “hierarchical visual processing” – possibly the wrong reference?

We thank the reviewers for pointing this out and we have adapted the sentence accordingly in line 68.

15. It is difficult to assess whether the imaged region actually corresponds to area A of PPC. The authors just located S1 and V1 using intrinsic imaging and defined the rest as PPC. Assuming the authors did not further pinpoint area A specifically, we would recommend just referring to the target area as PPC, not area A. Area A is relatively small and hard to hit without detailed mapping.

We agree with the reviewers that area A is relatively small and potentially difficult to target. As we have described in our response to reviewer 1 who had similar concerns, we perform intrinsic optical signal (IOS) imaging to determine the boundaries of V1 and the γ -barrel-column of S1 at the site of the cranial window (described in Methods lines 367 to 374) well before the imaging sessions. We have previously shown that the area between these boundaries corresponds to approximately area A of the PPC, using anatomical tracing methods (Mohan et al, 2018). By using the blood vessel patterns identified from the IOS as map, we restrict our field of view largely to area A of the PPC. It is well possible that we might also image the boundaries of areas RL and AM of the PPC. As such, we refer to the area that

we isolate with IOS as PPC, in line 373. At the end of the experiments mice are perfused and the brains recovered and sliced to confirm the location of the PPC, now stated in line 451.

16. Why does sound offset response start before the offset (Figure 1d) even using non-rooming sounds (Figure S2 b). It is probably incorrect to refer to these neurons as offset-responsive. See major comment 1 – unclear what is going on. Similarly, the neurons that are defined as whisker responsive start responding before the tactile onsets (Figure 1d, row 3 column 2).

In our reanalyzed data the sound offset response to both the looming and non-looming sounds start at the end of sound presentation. As we focus primarily on the whisker stimulus evoked responses, we have moved the sound and sound offset data to Supplementary Fig. 2b. Similarly the whisker-responsive neurons now start responding at the onset of the tactile stimulus, as shown in the revised Fig. 1d. We believe that these early onset responses were an artifact of the analysis procedures that we used in the initial analysis, that we now overcome with our revised analysis.

17. The number of S1 neurons that are responsive both in whisker and pairing sessions is surprisingly small (Figure 2c). This makes us suspect that substantial part of the response changes in pairing sessions could come from sensory adaptation.

The data in Fig. 2c is acquired from 4 mice, with 6 field views, with total number of 789 neurons. We agree with the reviewers that the number of neurons that are classified as responsive in both the whisker and pairing session is small. One explanation is that as we are imaging neurons that are close to the γ -barrel-column of S1 and not necessarily from the principle barrels that correspond to the multiple whiskers that we stimulate during these sessions. The interleaved paired-responsive neurons in S1 are stable in their response to the whisker stimulus, across the whisker, pairing and interleaved paired sessions (Fig. 2b bottom panel). If the changes in whisker stimulus response size observed in the transition from pairing to interleaved paired sessions in the whisker and pairing-responsive is due to sensory adaptation, this decrease should also be seen in the interleaved paired-responsive neurons. However this is not the case, arguing against such a possibility.

18. What is compared in Figure 3g?

Here we compare the mean response of each mismatch type (omission, decreased and increased) against the whisker stimulus response in the interleaved paired-responsive neurons for that corresponding interleaved session. We show that each mismatch response is larger than its corresponding interleaved paired trial response.

19. The DREADD experiments should be ideally performed with control experiment (e.g. CNO+ w/o receptor expression).

We agree with the reviewers that CNO control experiments without the DREADD expression are necessary to evaluate the role of M2 feedback using DREADD inhibition. As such we have performed these experiments and describe them in the revised Fig. 4 and Fig. 5.

20. Line 251 - 253: is this significant?

In our reanalyzed data we now have 45 mismatch-responsive neurons (out of 1528) in CNO+DREADD, compared to 78 mismatch-responsive neurons (out of 1371) in CNO control, and this difference is significant ($p = 7.24 \times 10^{-5}$).

21. The DREADD analysis in Figure 5 should probably be done on actual population averages (i.e. the entire population). First selecting responsive neurons and then plotting the responses of these subsets has the confound that the fraction of neurons in these subsets will change? May we be misunderstanding something here.

We agree with the reviewers on this aspect, and we now report the population mismatch size, as quantified over all the neurons in the revised Fig. 5. We show that the mismatch response is greatly reduced when M2 feedback is reduced via DREADD inhibition with CNO, as compared to the CNO control experiments without DREADD expression (Fig. 5c, 5d).

22. The argument in Figure S5 is hard to understand. The authors' argument reads like the introduction of mismatch increases top-down predictive inputs. Or did they want to say prediction was not formed completely during the pairing sessions?

We have now rephrased the sentence to say that the prediction is formed in the interleaved session, lines 308 to 309.

23. Are the serotypes and pseudotypes of the AAVs correct? We suspect the AAVs used were AAV2/1 (not AAV1/2 or AAV2).

We thank the reviewers for highlighting this error and we have corrected the revised manuscript accordingly.

RESPONSE TO REVIEWER COMMENTS

Reviewer #2 (Remarks to the Author):

After reading the revised submission, I am sorry to say that the manuscript still suffers from many of the same statistical errors and circular reasoning present in the original submission.

We thank the reviewer for the critical comments that have indeed highlighted a major error and oversight in our analysis. We have corrected this error and present our reanalyzed data below.

1) In regards to major point 1 in the original critique: The authors report that “3.7% of neurons (72 of 1923) can report the whisker stimulus omission in Fig.1”. Given that the alpha value for their Mann-Whitney test is 0.05, we would expect ~5% of neurons to have significant responses purely by chance, which is greater than the reported number of mismatch responses.

We re-verified our analysis and realized that some of the sessions from the mice were accounted for in replicates, in particular for the interleaved paired-responsive neurons. As a result, the total number of neurons reported was much higher than the actual values. We now report that the fraction of omission mismatch responsive neurons in Fig. 1 is 8.6% (72 of 834 neurons) and that of interleaved paired-responsive is 32.5% (271 of 834 neurons). These numbers for the mismatch responsive neurons are significantly above chance level. We apologize for this misunderstanding in the interpretation of our results.

In the rebuttal letter, they then start adding the results of different tests to claim that a larger percentage of the neurons respond to mismatch. However, if each statistical test has a 5% false positive rate, then we would expect running two statistical tests (e.g., for 0 ms delay and 1000 ms delay) and adding up the “significant” cells would yield ~10% responsive cells purely by chance, similar to their measured value of 12.9%. I remain unconvinced that the “mismatch” responses (that all subsequent analyses rely on) are not just a statistical artifact.

In Figure 2c (previously Figure 3c) we now report the fraction of omission mismatch responsive as 8.4% (79 of 938 neurons) with no delay, ie 0ms and as 10.0% (94 of 938 neurons) with a 1000ms delay. In Figure 2f (previously Figure 3f) we now report the fraction of omission mismatch responsive neurons as 12.9% (124 of 963 neurons), the fraction of decreased mismatch responsive neurons as 14.5% (140 of 963 neurons), and the fraction of increased mismatch neurons as 12.0% (116 of 963 neurons). We are thus convinced that these mismatch responses are not a statistical artifact.

2) In regards to major point 2 in the original critique (also pointed out by reviewer #3): Their reanalysis did not satisfactorily address the circular analysis issue. They continue to average responses from neurons that were determined a priori to be responsive during a particular epoch without cross-validation. This is clear in their figures, where a sharp band of elevated

activity can be seen running diagonally through the responses (Fig 1a, Fig 3a, c). As reviewer #3 pointed out, this is a classic example of circular analysis.

We now sort the average neuron responses in Fig. 1c and Fig. 2a and c (previously Fig.3a and c) according to the average response in the stimulus window, and the sharp band of elevated activity running diagonally is no longer present. No other changes (scale, thresholding etc) apart from an update in the neuron numbers for the interleaved paired-responsive neurons were done. We believe that the diagonal band of activity is purely due to the previous approach of sorting the neurons based on the time of their peak activity in the stimulus window and not on the selection criterion of being responsive for a given stimulus condition.

Later in the response letter, the authors state that they tried cross-validation, but that it led to slightly noisier responses (showing a single non-mismatch cell as an example). However, the slightly higher noise level in the responses is much preferable to plotting the responses of neurons preselected for responding during a particular interval. Plotting cross-validated responses is now standard procedure in the field.

We now show in Fig.1 for reviewer (below) the population averages of the various mismatch neurons from Fig.1d, Fig. 2c and f (previously Fig. 3c and f), where the neurons are selected using their average response based on either all the trials, or cross-validated where the selection was done using the odd trials and the responses were plotted using the even trials. It can be seen that the population averages are comparable when the neurons are selected on either criterion. As such, we have kept to our original approach in selecting the neurons based on their average response using all the trials.

Figure 1 for reviewer

3) In regards to major point 3 in the original critique (again pointed out by reviewer #3): The authors state that they fixed an error in the code that eliminated the acausal responses observed in the original manuscript. This mostly appears to be the case, but there do appear to still be “mismatch” responses occurring before the absent stimulus in Fig 5c.

The responses shown in Fig. 5c (now Fig. 4c) correspond to that of all recorded neurons, and not only mismatch responsive neurons. The average response of the mismatch responsive population is shown in Fig. 5a (now Fig. 4a). Hence the early response that starts during the presentation of the sound (as seen in the previous Fig. 5c) is likely due to sound responsive neurons. This is then followed by the mismatch response. These responses are

largely diminished by the chemogenetic suppression of M2 inputs to the PPC. We now discuss this in the discussion in lines 328-332 “Also, a response to the predicting sound in the population average of all neurons can be observed prior to the mismatch response (Fig. 4c). This is likely mediated by neurons responding to the predicting sound within the population, that are independent of mismatch neurons, and furthermore not recruited when M2 feedback is silenced.”

Reviewer #3 (Remarks to the Author):

The authors have done a commendable job addressing most of the concerns. There is one remaining concern that is still a bit worrying (major comment 2) – but I assume we are just misunderstanding something here. If this is indeed the case and the authors have a good explanation, we would recommend publication.

We thank the reviewer for the critical comments that have helped us to identify a major error and oversight in our data analysis. We have rectified this and present our re-analyzed data below.

This review was written in collaboration with a postdoc in the lab.

Major

1. Remove Figure 2. This adds very little in its current form (following Figure 1, what one would really want to know is if there are mismatch responses in S1, and they are conspicuously absent). More troubling, is that given that the authors likely have not regression to the mean corrected the analysis here (given that they chose not to do it elsewhere “because it looks too noisy”...)), this is almost certainly all a regression to the mean effect. If you select the neurons that are most responsive to stimulus A, these neurons will automatically look like they are less responsive to stimulus B and C, simply as a function of regression to the mean. How big this effect is depends on the trial-to-trial variability of the neurons. Thus, all this figure shows is that the trial-to-trial noise is likely higher in PPC than it is in S1. But most importantly, without a comparison of the mismatch responses between S1 and PPC it adds very little to the manuscript. Just remove it.

In Figure 2 for reviewer below, we show the average population responses of neurons that are responsive to the respective sessions, using cross-validation where the selection was done using the average from the odd trials and the responses were plotted using the even trials. With such an approach, it can still be observed that in the PPC selective groups of neurons are responsive mainly to their main session, in contrast to the responses in S1. With respect to mismatch responses in S1, we are convinced that more experiments are required before conclusions can be drawn from a comparison of mismatch responses between S1 and PPC. As such we agree with the reviewer and have removed Fig.2 from the manuscript. We now show the modulation of the pre-stimulus response of interleaved paired-responsive neurons with respect to pairing-responsive neurons in Fig. 1f.

Figure 2 for reviewer

2. Line 122. 3.7% of neurons were mismatch responsive. We don't understand, if $p < 0.05$ was used to define "responsive", any % of responsive neurons of 5% or less would mean the fraction of responsive neurons is at – or below – what one would expect to find by chance. I.e. 3.7% of mismatch responsive neurons would translate to finding "no evidence for mismatch responsive neurons". I assume we are misunderstanding something here.

As we have indicated in our reply to Reviewer 2, we realized on re-verifying our analysis that several sessions were counted in replicates, in particular the interleaved paired-responsive neurons. We have now corrected this error and report that 8.6% (72 of 834 neurons) of

neurons report the omission of the predicted stimulus. This fraction is above chance level. We apologize for this misunderstanding in the interpretation of our data.

Minor

3. The usage of “paired” and “pairing” in the figure legends (Figure 1e, Figure 2b,d,f,h, etc.) is still confusing. On top of this the authors use these terms to talk about neuron types and session types in the same figure (e.g. Fig 2 a, d, e & h). We would suggest to remedy this.

We have rectified this in the respective figure legends as suggested.

RESPONSE TO REVIEWER COMMENTS

Reviewer #2 (Remarks to the Author):

The second revision of their manuscript makes a few improvements on the previous version, but the authors appear to insist on using incorrect approaches for the analysis and display of their data.

We thank the reviewer for the critical comments and as suggested, we now present all our data with the cross-validated analysis, as we describe below.

1) In the last review, both reviewer #3 and I pointed out that there were fewer significant “prediction error” responsive cells (3.7%) than would be predicted by chance given the 0.05 alpha value used for their statistical tests. The authors found a mistake in their code that, once corrected, raised the number of significant cells to 8.6%. This is still a relatively small fraction of cells (~3.6% above that expected purely by chance), so there is some question of how prevalent these signals are in the population, but at least it is above the level expected by chance.

We now present our data with cross-validated responses, and we obtain a comparable number of mismatch responsive cells in Fig. 1 (71 out of 834, 8.5%) as before when the analysis was performed on cells selected based on the mean using all the trials. With cross-validation, we now report in Fig. 2c 8.2% of mismatch neurons (77 out of 938, for 0s and 1s delay) and in Fig. 2f we report 12.9% of omission mismatch neurons (125 out of 963), 14.5% of decreased mismatch neurons (140 out of 963) and 12.3% of increased mismatch neurons (118 out of 963). These fractions are comparable to our previous analysis and we remain convinced that these mismatch responses are not a statistical artifact, taking into account the cross-validated analysis that we have now employed as suggested.

I’m somewhat troubled that this is the second major mistake in their code that they’ve discovered during the review process (the other one caused “acausal” prediction error responses). Both of the errors had substantial effects on the direction and magnitude of the reported findings. Although I don’t have the time to review the code carefully, I am concerned by rigor of their analysis code if these major errors are still being uncovered (and likely would not have been discovered if they didn’t directly relate to reviewer critiques).

The errors in the neuron numbers were due to sessions being accounted for in replicates, due to inaccurate inputting of file paths. We re-verified all the paths for the analysis of our data and confirmed that there were no other errors. The “acausal” prediction error responses were a result of filtering that was part of the deconvolution process that we initially used. While the parameters for deconvolution could have been optimized, we have since chosen to instead present the neuronal responses as the original normalized calcium transients. We apologize

for the misunderstanding of our results due to the errors in our analysis, and we are appreciative of the reviewer comments that have highlighted our oversight in this aspect.

2) For reasons I fail to grasp, the authors are very resistant to using cross-validated responses, though this is clearly the correct way to analyze/plot the data (as pointed out by multiple reviewers). They showed us a few more example cells in the response to reviewers document, but continue to use the “circular” approach for displaying the data in the paper. For all analyses using “pre-selected” cells, the authors need to plot/analyze the cross-validated responses. This should be done throughout the paper, including in later Figures using similar selection criteria (e.g., Figure 4a-b).

We now present all our data using the cross-validated responses, including our supplementary data. Our overall findings are comparable to our original approach in analysis, and we have now moved the previous Figs. 1 to 4 to the supplementary section as Supplementary Figs. 5 to 8. We further detail the cross-validated analysis in the Methods section, in comparison to our original selection criteria.

3) I am still confused by the findings of Fig 4. It looks like the entire population response is reduced in the M2>PPC inhibition experiments, not specifically the mismatch response. Is it really accurate to say that “M2 contributes to the top-down prediction to the PPC”, specifically?

We thank the reviewer for highlighting this point. Indeed the response to the sound that predicts the whisker stimulus, as well the mismatch response is reduced when M2 inputs to the PPC are inhibited. Neurons that respond to the sound in the initial sound session (performed before the whisker session and presented in Supplementary Fig. 2a) no longer do so in the subsequent interleaved session. Hence it is possible that M2 can modulate the response to the predicting sound once the association is learned, either directly or indirectly via disinhibition. We now elaborate this in the Discussion, lines 279 to 287 (308 to 317 in the tracked manuscript). We further state in the Results, lines 220 to 222 (226 to 229 in the tracked manuscript), that “M2 can potentially contribute to the top-down prediction arriving at the PPC that modulates the response to the predicting sound, as well as the omission mismatch response”.

Reviewer #3 (Remarks to the Author):

We thank the reviewer for the critical comments and as suggested, we now present all our data with the cross-validated analysis, as we describe below.

A few minor remaining points:

1. The authors should really show cross-validated plots (as they are shown in response to review 2) instead of what they show in the paper. E.g. the very bizarre looking response to the 1s delay mismatch is not there anymore when using the proper cross validated way of plotting.

We now show cross-validated plots for all our data, including the supplementary figures.

2. I am not sure if the figure is new or different from previous submissions (I may have also missed in the previous review), but the difference in Figure 4c is likely not driven by the mismatch response? Primarily the sound response appears to be affected by M2 silencing.

In Fig. 4c where we now present the cross-validated plots, the mismatch response can be distinctly seen after the sound response that predicts the whisker stimulus. As we have described above in our comments to Reviewer 2, neurons that respond to the sound in the sound session (performed before the whisker session and presented in Supplementary Fig.2 a) no longer do so in the interleaved session, once the sensory association has been learned. This is unlike the neurons that respond to the whisker stimulus in the whisker, pairing and interleaved matched trials, and continue to do so across subsequent sessions with whisker stimuli. Hence it can be possible that M2 inputs to the PPC can modulate this response to the predictive sound, either directly or indirectly through disinhibition. This response is effectively suppressed when the M2 inputs are inhibited. We now discuss this in lines 279 to 287 of the Discussion (lines 308 to 317 in the tracked version of the manuscript).

3. I have pointed this out before, but the “paired” “pairing” terminology does not work. Figure labels of the form “post-stimulus interleaved paired – pairing” (as e.g. in Figure 3c – and shouldn’t panels c and e have different labels) are not intelligible.

We appreciate this point and we have replaced “paired” with “matched” throughout the whole manuscript.